



# How to perform global sensitivity analysis of a catchment-scale, distributed pesticide transfer model ? Application to the PESHMELBA model.

Emilie Rouzies [1], Claire Lauvernet [1], Bruno Sudret [2], and Arthur Vidard [3]

[1]INRAE, RiverLy, Lyon-Villeurbanne, 69625 Villeurbanne Cedex, France
[2]ETH Zurich, Chair of Risk, Safety and Uncertainty Quantification, Stefano-Franscini-Platz 5, CH-8093 Zurich, Switzerland
[3]Univ. Grenoble-Alpes, Inria, CNRS, Grenoble-INP, LJK, 38000 Grenoble, France

**Correspondence:** Emilie Rouzies (emilie.rouzies@inrae.fr)

**Abstract.** Pesticide transfers in agricultural catchments are responsible for diffuse but major risks to water quality. Spatialized pesticide transfer models are useful tools to assess the impact of the structure of the landscape on water quality. Before considering using these tools in operational contexts, quantifying their uncertainties is a preliminary necessary step. In this study, we explored how global sensitivity analysis can be applied to the recent PESHMELBA pesticide transfer model to quantify uncertainties on transfer simulations. We set up a virtual catchment based on a real one and we compared different approaches for sensitivity analysis that could handle the specificities of the model: high number of input parameters, limited size of sample due to computational cost and spatialized output. We compared Sobol' indices obtained from Polynomial Chaos Expansion, HSIC dependence measures and feature importance measures obtained from Random Forest surrogate model. Results showed the consistency of the different methods and they highlighted the relevance of Sobol' indices to capture interactions between parameters. Sensitivity indices were first computed for each landscape element (site sensitivity indices). Second, we proposed to aggregate them at the hillslope and the catchment scale in order to get a summary of the model sensitivity and a valuable insight into the model hydrodynamical behaviour. The methodology proposed in this paper may be extended to other modular and distributed hydrological models as there has been a growing interest in these methods in recent years.

## 1 Introduction

Pesticide transfers from fields to water bodies is a major but also complex environmental concern. Significant efforts are required to assess risks for aquatic and human lives. However, this is made difficult due to the complexity of processes at stake, the diversity and the fragmentation of agriculture landscapes where pesticides are applied (Campbell et al., 2004). Pesticide transfer models are essential tools to support risk management as they make simulation of contamination and transfers possible. They also make it possible to explore and compare alternative scenarios. However, there are many challenges for developing and using such models, highlighted for example in Gascuel-Odoux et al. (2009). In particular, to support decision-making, it is important to use physically based models that are simple enough to ensure flexibility (Reichenberger et al., 2007; Dosskey et al., 2011). A large range of models exist at a local scale (Adriaanse, 1997; Muñoz-Carpena et al., 1999; Carsel and Baldwin,





2000; Larsbo and Jarvis, 2003) but modelling approaches should also address the landscape scale. Indeed, the landscape configuration is of big influence on transfers and regulation and corrective actions can relevantly be set up at this scale. Building modular, process-based and distributed models is highly appropriate to address this scale and to take into account the landscape composition, (Buytaert et al., 2008; Kraft et al., 2012). This recent but promising approach is based on the coupling of different blocks representing one or several processes. This kind of models have flexible structures that are particularly promising for various risk assessment applications such as landscape management scenario exploration. Some models already exist following this idea based on the use of a modeling framework (Tortrat, 2005; Moussa et al., 2010; Branger et al., 2010; Rouzies et al., 2019). However, simulating at such scale also implies additional difficulties. First, catchment-scale models often require a high computational effort that cannot always be afforded (Herman et al., 2013). Second, spatialized process-based models may be characterized by a huge number of parameters that makes the parameter setting complex. Using such models thus raises new challenges about diagnosing model behaviour and uncertainty quantification (Gupta et al., 2008; van Griensven et al., 2006).

To address the issue of uncertainty quantification, Global Sensitivity Analysis (GSA) is a powerful tool that investigates how variations in the model outputs can be attributed to variations of its input factors (Saltelli et al., 2008; Pianosi et al., 2016). It aims at understanding the model complexity and formulating realistic assumptions for its use (Faivre et al., 2013). A large range of sensitivity analysis methods exist and they can be classified according to their objectives. Among others, *screening* methods identify (if there are) parameters that have a negligible influence on the output variability. Screening thus makes it possible to decrease the model complexity by setting constant values or removing non-influential parameters. *Ranking* methods classify input factors according to their relative contribution to output variability. In the specific case of process-based models, GSA may give an insight into a whole process representation (Gascuel-Odoux et al., 2009) and paves the way for potential improvements. Therefore, the GSA method should be chosen depending on the model characteristics and the analysis purpose (Song et al., 2015; Pianosi et al., 2016; Sarrazin et al., 2016). In the case of pesticide transfer modelling, variance decomposition and the Sobol' method (Sobol, 1993) have been widely used (e.g. Metta, 2007; Fox et al., 2010; Lauvernet and Muñoz-Carpena, 2018; Gatel et al., 2019). Although very popular, such methods have several limitations. First, they are characterized by a high computational cost although it can be alleviated using metamodelling techniques. For instance, computing polynomial chaos expansion (Ghanem and Spanos, 1991) directly gives Sobol' sensitivity indices at a low computational price (Sudret, 2008; Fajraoui et al., 2011; Wang et al., 2015). Second, from a methodological point of view, analysing the impact of an input factor through a variance indicator may constitute a *restrictive summary of the distribution* (Da Veiga, 2015). It can be especially unsatisfactory for highly-skewed or multimodal output variables (Liu et al., 2005; Borgonovo et al., 2011; Pianosi et al., 2016). In order to overcome such limitation, another category of methods describes the dependence between the output and each input factor from a probabilistic point of view (Székely et al., 2007; Da Veiga, 2015). In particular, in Da Veiga (2015), the Hilbert-Schmidt Independence Criterion (HSIC) proposed by Gretton et al. (2005b) is introduced to quantify the covariance between non-linear transformations of the input factor and the output variable. Although it has been used in only a few application fields so far (mainly related to risk assessment on nuclear accident contexts, see De Lozzo and Marrel 2016; Meynaoui 2019), the HSIC measure is promising and may be extended to hydrological and pesticide transfer modelling. Additionally, the growing interest for machine learning techniques is paving the way for new approaches of GSA, such as the Random Forest method





(RF). Indeed, its structure provides valuable information on feature importance that can be processed as sensitivity indices like in Harper et al. (2011) and Aulia et al. (2019) (see Antoniadis et al. (2021) for a review on the use of random forests for

sensitivity analysis).

Last but not least, pesticide transfer models are often fully spatialized, meaning that the interest area is divided into spatial units on which equations are solved locally. Such specificity should be carefully addressed to make the GSA step as informative and relevant as possible. Sensitivity can be examined at a local scale, on each spatial unit, which can make it computationally extensive. A second point of view consists in providing the user with synthetic measures that summarize the sensitivity over

the whole spatial domain. Saint-Geours (2012) introduces the notion of *site* sensitivity indices to refer to local GSA and the notion of *aggregated* sensitivity indices (sometimes called *block* sensitivity indices) to make reference to sensitivity indices with respect to all spatial points. Site sensitivity indices result in sensitivity maps that detail spatial contributions of influential parameters (Herman et al., 2013; Abily et al., 2016). Aggregated sensitivity indices are built either on a scalar objective function built from the spatialized output (like sum, average or maximum value) as explored in Saint-Geours et al. (2014) or

from a GSA performed on spatialized multidimensional output. In that case the extension of the GSA method to multidimensional output should be examined so as the interpretation of the results. For instance, the generalization of Sobol' indices to multidimensional output from Gamboa et al. (2013) is used in De Lozzo and Marrel (2016), to perform GSA on a spatialized radioactive material release model.

In this paper, we explore how the sensitivity of risk-assessment models can be relevantly tackled based on the example of the process-oriented, modular PESHMELBA model (PESticide and Hydrology: ModELling at the catchment scale). We perform a GSA of the PESHMELBA model and we aim at identifying relevant tools and a feasible methodology that could be transposed to other complex, distributed models. The spatialized aspect of the sensitivity analysis is particularly investigated. A broader scientific purpose is to determine how the information got from GSA can be used to better understand the processes that drive

transfers and fate of pesticides in the PESHMELBA model. The analysis is performed on a virtual scenario based on a real catchment in the Beaujolais region (France).

The paper is organized as follows: we describe the PESHMELBA model in Section 2.1 and the model setup in Section 2.2. Then, we introduce the different GSA methods used and the case study in Section 2.3 and the methodology used for landscape analysis in Section 2.4. Input sampling is described in Section 2.5. Results are presented in Section 3 focusing successively on

screening (Section 3.1), comparison of GSA methods (Section 3.2) and spatial analysis (Section 3.3).

## 2  Material and methods

### 2.1  The PESHMELBA model

The PESHMELBA model represents a catchment as a set of interconnected components that stand for landscape elements such as plots, Vegetative Filter Strips (VFSs), ditches, hedges or rivers (Rouzies et al., 2019). In order to respect the spatial

organization and the heterogeneity of the landscape, it deals with mesh elements that can be surfaces or lines. Surface mesh





elements are called Homogeneous Units (HUs). A HU is a portion of landscape that is homogeneous in terms of hydrodynamical processes and agricultural practices. Linear mesh elements are called reaches. A reach is characterized by its nature (so far ditch, river or hedge) and by its neighbouring components: it is at most in contact with one elementary mesh element on each bank. In addition to its geometric or hydrodynamic properties, each mesh element is characterized by its one-way connections

with the neighbouring components that stand at a lower altitude. One or several processes are represented on each element depending on its nature. Lateral transfers at surface and in subsurface between elements are also integrated. Independent codes called units are used to simulate the different processes, depending on the knowledge the user has on the targeted catchment functioning. Then, the OpenPALM coupler (Fouilloux and Piacentini, 1999; Buis et al., 2006) is used to couple the units and to build the complete application. OpenPALM has adapted features to easily deal with spatial and temporal aspects of the model.

For example, synchronization tools make it possible to couple processes with different time steps. The final structure is highly modular and process representations can easily be added, upgraded or removed depending on the landscape description. These features make PESHMELBA particularly suitable for scenario exploration.

PESHMELBA focuses on surface and subsurface transfers of water and pesticides. An extensive description of elements and processes already included can be found in Rouzies et al. (2019). The PESHMELBA version used in this study integrates a

representation of water and pesticide transfers on plots, VFSs and rivers. Each plot or VFS is represented by a unique column of soil divided into vertical cells. In such a column, vertical infiltration is simulated using a solution of the 1D Richards equation proposed by Ross (Ross, 2003). An adapted set of parameters makes it possible to represent high infiltration rate, surface runoff reduction and enhanced adsorption and degradation on VFSs. Root-water uptake is integrated based on Varado et al. (2006). Surface runoff routing is represented based on the kinematic wave (Lighthill and Whitham, 1955) and the Darcy law

(Darcy, 1857) is used for lateral subsurface transfers. In addition to shallow groundwater tables, PESHMELBA also represents shallowly perched water tables and associated lateral transfers. Finally, reactive transfer of solutes is represented: advection, degradation based on a first order law and adsorption, based on linear or Freundlich isotherms are integrated. Each river or ditch reach is represented by a unique tank. The River1D module (Branger et al., 2010) solves the kinematic wave equation for water routing and pesticide advection in the network. Groundwater-river exchanges are represented by the Miles formulation

adapted by Dehotin et al. (2008).

## 2.2 Model setup

A virtual scenario of limited size was set from a portion of la Morcille catchment (France) in order to explore different GSA methods and to ease interpretation of spatialized results. The chosen portion was selected so as to remain representative of the global composition of La Morcille catchment in terms of soil, slope, type and size of elements as well as interface length

between them. The chosen scenario was composed of 10 vineyard plots, 4 vegetative filter strips and 6 river reaches that delimit a left and a right slope (see Figure 1).

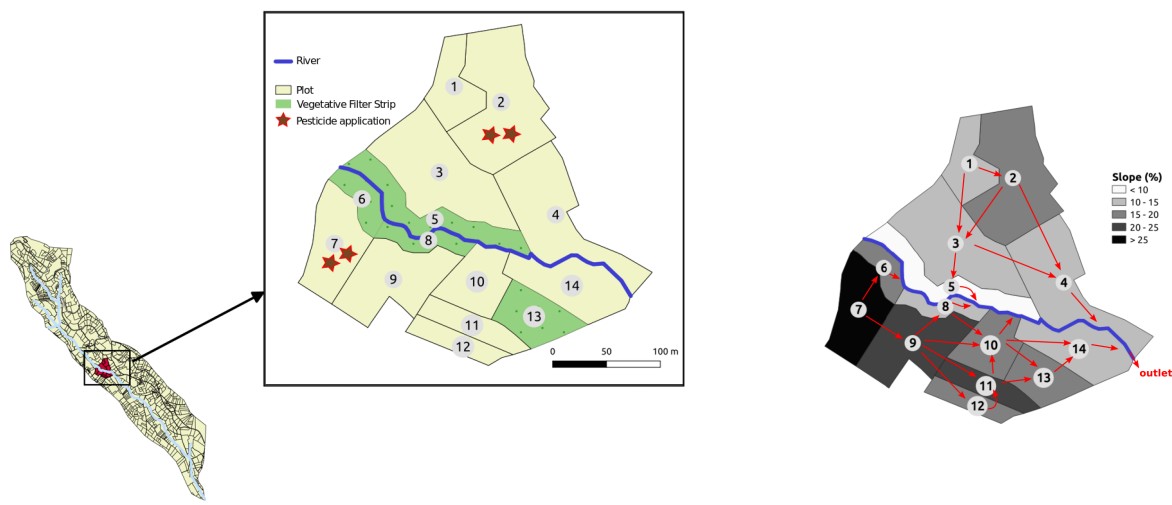

**Figure 1.** Left: portion of La Morcille catchment selected to perform sensitivity analysis. Yellow units stand for vineyard plots while green units stand for vegetative filter strips. Brown stars denote locations of pesticide application. Right: slopes and connections between elements.

Soils types on the catchment are mainly sandy (Peyrard et al., 2016). We used the classification from Frésard (2010) that groups soil types into 3 main Soil Units (SUs). Each SU is defined by the vertical succession of 3 or 4 soil layers, also called soil horizons: one surface horizon, 1 or 2 intermediary horizons and one deep horizon as depicted in Figure 2. Note that interface depths can vary from one SU to another. The reader may refer to Rouzies et al. (2019) for further details on how soil types 125 and soil horizons are represented in PESHMELBA. At the catchment scale, the classification resulted in the following SU (see Figure 2): sandy soil (SU1), sandy soil on clay on the right bank plateau (SU2) and heterogeneous sandy soils on lower slopes and thalwegs (SU3).

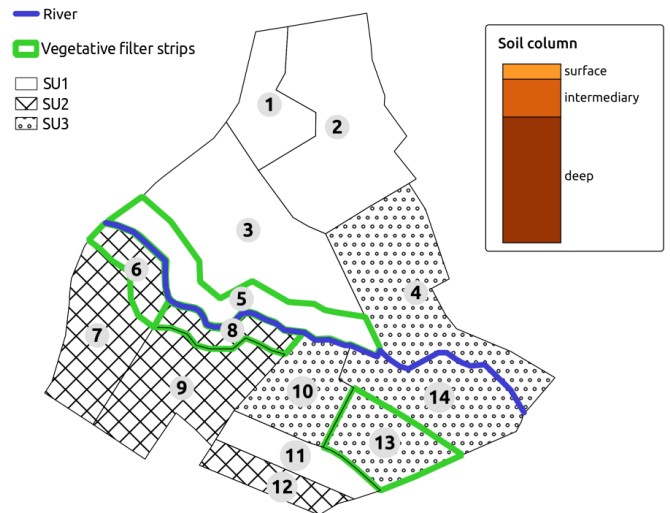

**Figure 2.** Soil type locations for the case study. Green contours show the vegetative filter strips.





Spatial arrangement was set in order to be as realistic as possible in terms of possible interfaces between SUs. Each SU
was set at least on one vineyard plot and one VFS on the virtual scenario. Bulk density and organic carbon content data were
available from Van den Bogaert (2011) and Randriambololohasinirina (2012). Hydrodynamic parameters for each soil type are
described in Table 1. Retention values measured by Van den Bogaert (2011) were used to fit retention curve using SWRCfit
tool (Seki, 2007). A Schaap-Van Genuchten conductivity curve was used (Schaap and van Genuchten, 2006; Ross, 2006)
whose matching point at saturation $Ko$ and empirical pore-connectivity $L$ were derived from conductivity data and retention
parameters from retention curve fitting. Surface organic carbon content was set equal to that of the first soil horizon on plots
and VFSs. For each SU, only the first soil horizon on VFSs differs from vineyard plots so as to highlight enhanced infiltration
capacities. Surface horizon on VFSs was characterized by a 2.8%-organic carbon content (Randriambololohasinirina, 2012)
and a saturated hydraulic conductivity of 150 mm.h$^{-1}$ (or $4.31 \cdot 10^{-5}$ m.s$^{-1}$) following Catalogne et al. (2018). In the absence
of data to characterize potential anisotropy of vertical and horizontal saturated conductivities $Ks_v$ and $Ks_h$, isotropy was
considered, thus the ratio $Ks_h/Ks_v$ was set to 1 on the catchment.

|  | Horizon | depth | *thetas* | *thetar* | *hg* | *n* | *Ks* | *Ko* | *L* |
|  |  | [m] | [m$^3$m$^{-3}$] | [m$^3$m$^{-3}$] | [m] | [-] | [ms$^{-1}$] | [ms$^{-1}$] | [-] |
| SU1 | 11-14 | 0.05 | 0.34 | 0.04 | $-9.69 \cdot 10^{-2}$ | 1.27 | $3.93 \cdot 10^{-5}$-$4.31 \cdot 10^{-5}$ | $2.86 \cdot 10^{-7}$ | -8.43 |
|  | 2 | 0.5 | 0.34 | 0.05 | $-3.29 \cdot 10^{-2}$ | 1.20 | $8.64 \cdot 10^{-5}$ | $2.28 \cdot 10^{-7}$ | -6.52 |
|  | 3 | 0.65 | 0.32 | 0.08 | $-2.09 \cdot 10^{-2}$ | 1.20 | $5.39 \cdot 10^{-5}$ | $7.47 \cdot 10^{-7}$ | -4.24 |
|  | 4 | 4 | 0.28 | 0.07 | $-5.99 \cdot 10^{-2}$ | 1.23 | $3.11 \cdot 10^{-5}$ | $1.47 \cdot 10^{-6}$ | -0.14 |
| SU2 | 12-15 | 0.035 | 0.34 | 0.04 | $-9.69 \cdot 10^{-2}$ | 1.27 | $3.93 \cdot 10^{-5}$-$4.31 \cdot 10^{-5}$ | $2.86 \cdot 10^{-7}$ | -8.43 |
|  | 6 | 0.4 | 0.35 | 0 | $-6.60 \cdot 10^{-2}$ | 1.13 | $2.16 \cdot 10^{-5}$ | $3.19 \cdot 10^{-7}$ | 9.66 |
|  | 7 | 0.55 | 0.32 | 0 | $-7.18 \cdot 10^{-2}$ | 1.08 | $9.60 \cdot 10^{-6}$ | $1.67 \cdot 10^{-7}$ | -10 |
|  | 8 | 4 | 0.42 | 0 | $-0.30 \cdot 10^{-2}$ | 1.08 | $3.98 \cdot 10^{-6}$ | $9.72 \cdot 10^{-8}$ | 10 |
| SU3 | 13-16 | 0.06 | 0.34 | 0.04 | $-9.69 \cdot 10^{-2}$ | 1.27 | $3.93 \cdot 10^{-5}$-$4.31 \cdot 10^{-5}$ | $2.86 \cdot 10^{-7}$ | -8.43 |
|  | 9 | 0.45 | 0.33 | 0.08 | $-6.72 \cdot 10^{-2}$ | 1.26 | $3.05 \cdot 10^{-5}$ | $3.36 \cdot 10^{-7}$ | 0.42 |
|  | 10 | 4 | 0.32 | 0.06 | $-3.56 \cdot 10^{-2}$ | 1.18 | $2.38 \cdot 10^{-5}$ | $3 \cdot 10^{-7}$ | 1.05 |

**Table 1.** Hydrodynamics parameters for SU1, 2 and 3 based on Van Genuchten retention curve and on Schaap-Van Genuchten conductivity
curve fitting. Parameters are described in Table 2. Values for the surface horizon are explicitly distinguished between plot and VFS when
they are different. Horizons 11, 12, 13 are surface horizons for plots whereas horizons 14, 15, 16 are surface horizons for VFSs.

The pesticide chosen in this study is the tebuconazole as it is a fungicide widely used on la Morcille catchment. It is a
slightly mobile molecule and we used a Freundlich isotherm to describe its adsorption equilibrium. Adsorption parameters were
obtained from Lewis et al. (2016) ($Koc$ = 769 mL.g$^{-1}$, Freundlich isotherm exponent = 0.84). Surface degradation coefficient
was also taken from Lewis et al. (2016) ($DT50$ = 47.1 days) and a decreasing degradation rate in function of depth was set
as in FOCUS (2001). A 500g-application was considered at the beginning of the simulation on plots 2 and 7 (see Figure 1).

Most of the transformation and adsorption of tebuconazole was supposed to happen on plots and VFSs at this modelling scale. Therefore, no adsorption or degradation was simulated in the river.

A no-flux boundary condition was applied on all sides except on surface where rain and potential evapotranspiration were considered. Rain data were extracted from BDOH database (Gouy et al., 2015). A 3-month simulation was performed on
a winter period characterized by long and intense rain events (670 mm cumulated). Potential evapotranspiration (PET) data were obtained from MeteoFrance for the neighbouring site of Liergues (MeteoFrance, 2008). Data were averaged over 10-day periods and corrected by a factor -11 % to match La Morcille site (Durand, 2014; Caisson, 2019). Rain and PET data for the simulation are summarized in Figure 3.

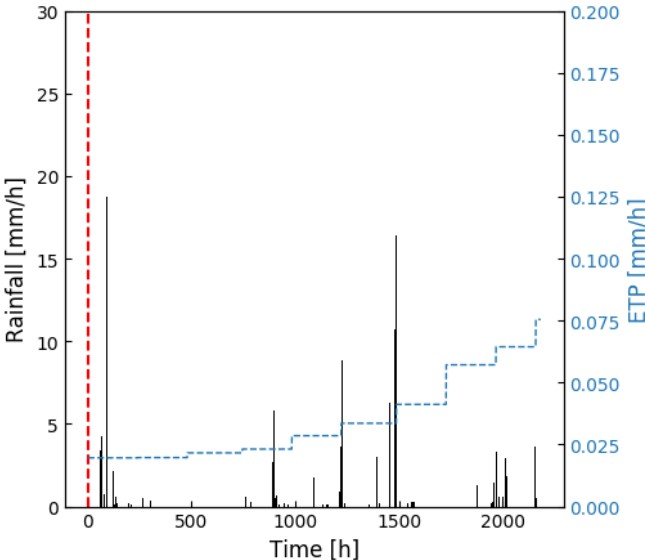

**Figure 3.** Climatic forcing (rain and potential evapotranspiration) for the simulation. The dotted red line stands for the one-shot pesticide application.

Although virtual, we aimed at setting initial conditions as plausible as possible for this scenario. Initial water table levels
were deduced from piezometric data on a neighbouring hillslope. Data from several piezometers were available on a transect, perpendicular to the river. Data were extrapolated over the virtual hillslope width on both sides of the river. All soil columns were supposed to be in hydrostatic equilibrium at the beginning of the simulation. An upstream 0.177 $m^3s^{-1}$-flow was considered in the river based on local measurements (Gouy et al., 2015).

Two types of vegetation were represented in this scenario. Vineyard cover was considered on plots while permanent grassland
was simulated on VFSs. Considering the period of simulation (3 months), a fixed root depth ($Zr$=2.62 m) and a fixed root density in the first 10 % of the root depth ($F10$=37 %) were considered for vineyards following values reported in Smart et al. (2006) and confirmed by expert knowledge in the area. The root depth ($Zr$) was set to 0.9 m and the root density in the first 10 % of the root depth ($F10$) was set to 33.5 % for grassland (Brown et al., 2007). For vineyards, the Leaf Area Index (LAI) was





assumed to increase from leaves formation until a maximum value before declining until harvest. Dates and associated values

for this development cycle were taken from Brown et al. (2007). They are summarized in Appendix A1 so as the constant
LAI value set for grassland on VFSs, based on Brown et al. (2007). The remaining parameters for root extraction model were
fixed to nominal values proposed by Varado et al. (2006); Li et al. (2001). Manning coefficients were set from data reported in
Arcement and Schneider (1989). A mature row crop value (0.033 s.m$^{-1/3}$) was chosen for vineyard while a high grass pasture
value (0.2 s.m$^{-1/3}$) was set for VFS cover.

Finally, ponding height was set to 0.01 m on vineyard plots while an increased value was set on VFSs (0.05 m). According
to Gao et al. (2004); Walter et al. (2007), the surface mixing layer thickness was set to 0.01 m on both plot and VFS domains.
In the river, the distance between the river bed and the limit of impervious saturated zone ($di$) was set to 1.5 m (ERT field
measure, personal communication) and the saturated hydraulic conductivity ($Ks$) was set to 2.38·10$^{-5}$ ms$^{-1}$ accordingly to
local saturated conductivities in the neighbouring soil.

## 2.3 Global Sensitivity Analysis methods

We explored several Global Sensitivity Analysis methods in order to identify the most suitable approach to deal with the
specificities of the targeted application: heterogeneous and coupled structure of the model, highly non linear processes and
spatialized aspect. In what follows, we denote $Y \in \mathbb{R}$ a given model output. Y is function of a multivariate input random vector
$\underline{\mathbf{X}} = (X_1, ..., X_M) \in \mathbb{R}^M$ such as $Y = \mathcal{M}(X_1, ..., X_M)$, with $\mathcal{M}$ the computational model that maps $\underline{\mathbf{X}}$ onto the output space.

The following three sections detail the mathematical theory under each method investigated to compute sensitivity indices for
the PESHMELBA model: the Sobol' indices based on variance decomposition using generalized polynomial chaos expansion,
the HSIC dependence measures and the feature importance measures obtained from Random Forest metamodelling technique.
For each method, extension to multidimensional output $\underline{\mathbf{Y}} \in \mathbb{R}^d$ is also investigated. Readers in a hurry may directly jump to
Section 2.4 that provides a qualitative summary and the technical details on the implementation for this case study.

### 2.3.1 Variance decomposition

**Definition**

Variance-based methods aim at determining how input factors contribute to the output variance (Faivre et al., 2013). One of
the most popular variance-based method is the Sobol' method (Sobol, 1993). It is based on the decomposition of the output
variable Y into summands of increasing dimension:

$$Y = \mathcal{M}(x_1, ..., x_M) = m_0 + \sum_{s=1}^{M} \sum_{i_1 < ... < i_s}^{M} m_{i_1, ... i_s}(x_{i_1}, ..., x_{i_s}), \qquad (1)$$

where $1 \le i_1 < ... < i_s \le M$ and where summands $m_{i_1, ... i_s}(x_{i_1}, ..., x_{i_s})$ are defined so that their integral over any of its in-
dependent variables is zero. This last property implies that the summands are orthogonal, which makes this decomposition
unique. It is referred to as ANOVA for ANalysis Of VAriances (Archer et al., 1997). If $Y = \mathcal{M}(\underline{\mathbf{X}})$ is square integrable, all





summands $m_{i_1,...,i_s}$ are also square integrable and one can square and integrate Eq. (1) leading to:

$$\int \mathcal{M}^2(\underline{\mathbf{X}})\mathrm{d}\underline{\mathbf{X}} = m_0^2 + \sum_{s=1}^{M} \sum_{i_1<...<i_s}^{M} \int m_{i_1,...,i_s}^2 \mathrm{d}x_{i_1}...\mathrm{d}x_{i_s}. \tag{2}$$

$\int \mathcal{M}^2(\underline{\mathbf{X}})\mathrm{d}\underline{\mathbf{X}} - m_0^2$ is the total variance of $\mathcal{M}(\underline{\mathbf{X}})$ denoted $\mathrm{Var}[Y]$ in what follows. If one introduces the notion of partial variances $\mathrm{V}_{i_1,...,i_s}$ so that:

$$\mathrm{V}_{i_1...i_s} = \int m_{i_1,...,i_s}^2(x_{i_1},...x_{i_s})\mathrm{d}x_{i_1},...,\mathrm{d}x_{i_s} \text{ for } 1 \le i_1 < ... < i_s \le M, s = 1,...,M, \tag{3}$$

Then, Eq. (2) reads:

$$\mathrm{Var}[Y] = \sum_{s=1}^{M} \sum_{i_1<...<i_s}^{M} \mathrm{V}_{i_1,...,i_s}. \tag{4}$$

In such formulation, $\mathrm{V}_{i_1...i_s}$ indicates the portion of variance that can be attributed to interactions between input parameters $X_i, i \in i_1,...i_s$. Single-index variance $\mathrm{V}_i$ indicates the portion of variance that can be attributed to the main effect of $X_i$ taken alone. From the above, one can define Sobol' indices as:

$$\mathrm{S}_{i_1,...,i_s} = \frac{\mathrm{V}_{i_1...i_s}}{\mathrm{Var}[Y]}. \tag{5}$$

By definition, $0 \le \mathrm{S}_{i_1,...,i_s} \le 1$. In particular, first order sensitivity indices $\mathrm{S}_i = \frac{\mathrm{V}_i}{\mathrm{Var}[Y]}$ only account for main effects. These indices are usually calculated as a first step as they often account for a large portion of the variance (Faivre et al., 2013). Total sensitivity indices $\mathrm{S}_{\mathrm{T}_i}$ evaluate the total effect of an input factor $X_i$ on the output by taking into account its main effect $\mathrm{S}_i$ and all interaction terms that involve it:

$$\mathrm{S}_{\mathrm{T}_i} = \sum_{\mathcal{I}_i} \mathrm{S}_{i_1,...,i_s}, \quad \mathcal{I}_i = \{(i_1,...,i_s) \mid \exists k, 1 \le k \le s, i_k = i\}. \tag{6}$$

### 210 Sobol' indices computation using generalized polynomial chaos expansion

In the original paper, Sobol (1993) uses Monte Carlo simulations to compute these sensitivity indices. This method is based on two Monte Carlo samples of size $N$. However, such estimation of sensitivity indices requires $2^N$ estimations of the model. Saltelli et al. (2008) recommend to choose $N = k(M+2)$ where $M$ is the number of input factors and $k$ a number between 500 and 1000. Using such a sample size may end up being unfeasible in case of computationally expensive models. The use

of a metamodel is then an interesting alternative, in particular with the Polynomial Chaos Expansions (PCE) that have the advantage of providing direct estimation of Sobol' indices. PCE theory is briefly introduced in what follows, mainly based on Le Gratiet et al. (2017).

Polynomial Chaos Expansion metamodelling technique is based on homogeneous chaos theory (Wiener, 1938). It provides

a functional approximation of the computational model based on the projection of the model output on a suitable basis of





stochastic polynomial functions in the random inputs (Ghanem and Spanos, 1991). For any square integrable output random variable $Y = \mathcal{M}(\underline{X})$, $Y \in \mathbb{R}$, its polynomial chaos expansion is expressed as follows:

$$Y = \sum_{\boldsymbol{\alpha} \in \mathbb{N}^M} \gamma_{\boldsymbol{\alpha}} \Psi_{\boldsymbol{\alpha}}(\underline{X}), \tag{7}$$

where the $\Psi_{\boldsymbol{\alpha}}$'s are multivariate orthonormal polynomials that constitute the basis and $\gamma_{\boldsymbol{\alpha}}$'s the associated coordinates. Multi-
variate polynomials are built as tensor products of univariate polynomials. For example, in the case of a multivariate polynomial of degree $\boldsymbol{\alpha}$, one defines the multi-index $\boldsymbol{\alpha} = (\alpha_1, ..., \alpha_M)$ with $|\boldsymbol{\alpha}| = \sum_{i=1}^{M} \alpha_i$ and $\Psi_{\boldsymbol{\alpha}}$ is expressed as:

$$\Psi_{\boldsymbol{\alpha}}(\mathbf{X}) = \prod_{i=1}^{M} \Psi_{\alpha_i}(x_i), \tag{8}$$

where $\Psi_{\alpha_i}(x_i)$'s is the univariate polynomial of degree $\alpha_i$ from the orthonormal family associated to variable $x_i$. Univari-
ate polynomials are chosen among classical families of polynomials, depending on the input factor distributions (Xiu and
Karniadakis, 2002). Expansion from Eq. (7) is usually truncated to a finite sum for practical computation:

$$Y \approx \sum_{\boldsymbol{\alpha} \in \mathcal{A}} \gamma_{\boldsymbol{\alpha}} \Psi_{\boldsymbol{\alpha}}(\underline{X}), \tag{9}$$

where $\mathcal{A} \subset \mathbb{N}^M$ is the subset of selected multi-indices of multivariate polynomials. Different truncation schemes exist, the most obvious being the standard truncation scheme that only keeps multivariate polynomials of total degree less or equal than $p$:
$\mathcal{A}^{M,p} = \{\boldsymbol{\alpha} \in \mathbb{N}^M : |\boldsymbol{\alpha}| \leq p\}$.


PCE truncated decomposition can be rearranged into summands of increasing order. Such formulation is based on sets $\mathcal{A}_{i_1,...,i_s}$ that contain all $\boldsymbol{\alpha} = (\alpha_1, ..., \alpha_M)$ tuples such that only indices $(i_1, ..., i_s)$ are nonzero:

$$\mathcal{A}_{i_1,...,i_s} = \left\{ \boldsymbol{\alpha} : \begin{array}{l} \alpha_k > 0, \forall k = 1, ..., M \mid k \in (i_1, ..., i_s) \\ \alpha_k = 0, \forall k = 1, ..., M \mid k \notin (i_1, ...i_s) \end{array} \right\} \tag{10}$$

As detailed in Sudret (2008), the truncated expansion in Eq. (9) can be written as:

$$Y = \gamma_0 + \sum_{i=1}^{M} \sum_{\boldsymbol{\alpha} \in \mathcal{A}_i} \gamma_{\boldsymbol{\alpha}} \psi_{\boldsymbol{\alpha}}(x_i) + \sum_{1 \leq i_1 < i_2 \leq M} \sum_{\boldsymbol{\alpha} \in \mathcal{A}_{i_1,i_2}} \gamma_{\boldsymbol{\alpha}} \psi_{\boldsymbol{\alpha}}(x_{i_1}, x_{i_2}) + ... +$$
$$\sum_{1 \leq i_1 < ... < i_s \leq M} \sum_{\boldsymbol{\alpha} \in \mathcal{A}_{i_1,...,i_s}} \gamma_{\boldsymbol{\alpha}} \psi_{\boldsymbol{\alpha}}(x_{i_1}, ..., x_{i_s}) + ... +$$
$$\sum_{\boldsymbol{\alpha} \in \mathcal{A}_{1,2,...,M}} \gamma_{\boldsymbol{\alpha}} \psi_{\boldsymbol{\alpha}}(x_1, ..., x_M). \tag{11}$$


By unicity of the ANOVA decomposition, Eq. (11) is therefore the ANOVA of Y. As a result, Sobol' indices can be deduced from gathering PCE coefficients according to the dependency on each basis polynomial. Because of the orthonormality of the




multivariate polynomials, the partial variance are nothing but the sum of the squares of the PCE coefficients. After square-summing and normalizing, one gets:

$$\mathrm{S}_{i_1,\ldots,i_s} = \sum_{\boldsymbol{\alpha} \in \mathcal{A}_{i_1,\ldots,i_s}} \frac{\gamma_{\boldsymbol{\alpha}}^2}{\mathrm{Var}[Y]} \tag{12}$$

where $\mathrm{Var}[Y] = \sum_{\boldsymbol{\alpha} \in \mathcal{A}} \gamma_{\boldsymbol{\alpha}}^2$ is the total variance. In particular, first-order indices read:

$$\mathrm{S}_i = \sum_{\boldsymbol{\alpha} \in \mathcal{A}_i} \frac{\gamma_{\boldsymbol{\alpha}}^2}{\mathrm{Var}[Y]}, \quad \mathcal{A}_i = \{\boldsymbol{\alpha} \in \mathbb{N}^M \mid \alpha_i > 0, \alpha_j = 0, \forall j \neq i\} \tag{13}$$

and total-order indices are expressed as:

$$S_i^T = \sum_{\boldsymbol{\alpha} \in \mathcal{A}_i^T} \frac{\gamma_{\boldsymbol{\alpha}}^2}{\mathrm{Var}[Y]}, \quad \mathcal{A}_i^T = \{\boldsymbol{\alpha} \in \mathbb{N}^M \mid \alpha_i > 0\}. \tag{14}$$

**Extension to multidimensional outputs**

In this paper, Sobol' indices for multidimensional outputs are calculated following the formulation by Gamboa et al. (2013) for generalized Sobol' indices. Denoting the output variable $\underline{\mathbf{Y}} = \mathcal{M}(\underline{\mathbf{X}}) \in \mathbb{R}^d$ such as $\mathbb{E}[||\underline{\mathbf{Y}}||]^2 < \infty$, for any input parameter $X_i$, generalized first order Sobol' indices are based on the Hoeffding decomposition of the model function $\mathcal{M}$:

$$\mathcal{M}(\underline{\mathbf{X}}) = c + \mathcal{M}_i(\underline{\mathbf{X}}_i) + \mathcal{M}_{\sim i}(\underline{\mathbf{X}}_{\sim i}) + \mathcal{M}_{i,\sim i}(\underline{\mathbf{X}}_i, \underline{\mathbf{X}}_{\sim i}), \tag{15}$$

where $c = \mathbb{E}[\underline{\mathbf{Y}}]$, $\mathcal{M}_i = \mathbb{E}[\underline{\mathbf{Y}}|\underline{\mathbf{X}}_i] - c$, $\mathcal{M}_{\sim i} = \mathbb{E}[\underline{\mathbf{Y}}|\underline{\mathbf{X}}_{\sim i}] - c$ and $\mathcal{M}_{i,\sim i} = \underline{\mathbf{Y}} - \mathcal{M}_{\sim i} - \mathcal{M}_i - c$ where $\sim i$ is the subset $\{1 \leq k \leq M, k \neq i\}$. After computing the covariance and scalarizing both sides of Eq. (15) by applying the trace operator $\mathrm{Tr}$, one gets:

$$\mathrm{Tr}(\Sigma) = \mathrm{Tr}(C_i) + \mathrm{Tr}(C_{\sim i}) + \mathrm{Tr}(C_{i,\sim i}) \tag{16}$$

where $\Sigma$, $C_i$, $C_{\sim i}$ and $C_{i,\sim i}$ respectively denote the covariance matrices of $\underline{\mathbf{Y}}$, $\mathcal{M}_i(\underline{\mathbf{X}}_i)$, $\mathcal{M}_{\sim i}(\underline{\mathbf{X}}_{\sim i})$ and $\mathcal{M}_{i,\sim i}(\underline{\mathbf{X}}_i, \underline{\mathbf{X}}_{\sim i})$.
As soon as $\underline{\mathbf{Y}}$ is not constant, one can divide both sides of Eq. (16) by $\mathrm{Tr}(\Sigma)$, leading to:

$$S_i = \frac{\mathrm{Tr}(C_i)}{\mathrm{Tr}(\Sigma)} = \frac{\sum_{j=1}^d \mathrm{Var}[Y_j] S_{i,j}}{\sum_{j=1}^d \mathrm{Var}[Y_j]}, \tag{17}$$

where $\mathrm{Var}[Y_j]$ stands for the variance of the scalar $j^{\mathrm{th}}$ component of $\underline{\mathbf{Y}}$ and $S_{i,j}$ is the first order Sobol' indice of $Y_j$ associated to $X_i$. The Hoeffding decomposition can be generalized to any subset $u \in \{1,..,M\}$ and Sobol' indices at any order can be then computed the same way.

**2.3.2 HSIC dependence measure**

The following sections briefly describe the HSIC dependence measure and introduce a method for computing an estimator. Although the description below considers 1d-input factor and 1d-output, the HSIC theory can be easily extended to multidimensional variables (see De Lozzo and Marrel 2016 for example).





The HSIC theory relies on Reproducing Kernel Hilbert Space (RKHS) and kernel functions. The Hilbert space $\mathcal{H}$ of functions from $\mathcal{X}$ to $\mathbb{R}$ associated to the scalar product $\langle \cdot, \cdot \rangle_{\mathcal{H}}$ is a RKHS if for all $x \in \mathcal{X}$, the application $h \in \mathcal{H} \mapsto h(x)$ is a continuous linear form. The Riesz representation theorem stands that to each value $x$ in the space $\mathcal{X}$ corresponds a unique function $\phi_x \in \mathcal{H}$ so that $\forall h \in \mathcal{H}, h(x) = \langle h, \phi_x \rangle_{\mathcal{H}}$. Such RKHS thus provides us with a mapping from $\mathcal{X}$ to $\mathcal{H}$. This mapping and thus the RKHS $\mathcal{H}$ is defined by its unique kernel function $k$ such that $\forall (x, x') \in \mathcal{X}^2, \langle \phi_x, \phi_{x'} \rangle_{\mathcal{H}} = k(x, x')$

**Definition**

Let $\mathcal{F}_i$ denote the RKHS composed of all continuous bounded functions of input $X_i$ with values in $\mathbb{R}$ and $\mathcal{G}$ the RKHS composed of real-valued continuous bounded functions of output $Y$ with values in $\mathbb{R}$. $\langle \cdot, \cdot \rangle_{\mathcal{F}_i}$ (resp. $\langle \cdot, \cdot \rangle_{\mathcal{G}}$) is the inner product on $\mathcal{F}_i$ (resp. $\mathcal{G}$) and $k_{X_i}$ (resp. $k_Y$) is the corresponding kernel function.

The main idea behind the Hilbert-Schmidt Independence Criterion (HSIC) used for GSA is to calculate the cross-correlation between any non-linear transformations of some input factor $X_i$ and the output $Y$ (De Lozzo and Marrel, 2016). It means that such dependence measure simultaneously captures a very broad spectrum of forms of dependency between the variables (Meynaoui et al., 2018). Such non-linear transformations of $X_i$ and $Y$ are described by the elements of $\mathcal{F}_i$ and $\mathcal{G}$. The HSIC measure then corresponds to the cross-covariance between any functions $f$ and $g$ in these RKHS, namely $\text{cov}(f(X_i), g(Y))$. To do so, one may define the cross-covariance operator $C[\mathcal{GF}_i] : \mathcal{G} \to \mathcal{F}_i$ which is the unique operator from $\mathcal{G}$ to $\mathcal{F}_i$ such that:

$$\forall f_i \in \mathcal{F}_i, \forall g \in \mathcal{G}, \langle f, C[\mathcal{GF}_i](g) \rangle_{\mathcal{F}_i} = \text{cov}(f_i(X_i), g(Y)). \tag{18}$$

Finally, the HSIC measure corresponds to the square of the Hilbert-Schmidt norm of the cross-correlation operator, which is:

$$HSIC(X_i, Y)_{\mathcal{F}_i, \mathcal{G}} = ||C[\mathcal{GF}_i]||_{HS}^2 = \sum_{j,k} \langle u_j^i, C[\mathcal{GF}_i](v_k) \rangle_{\mathcal{F}_i} = \sum_{j,k} \text{cov}(u_j^i(X_i), v_k(Y)), \tag{19}$$

where $(u_j^i)_{j \geq 0}$ and $(v_k)_{k \geq 0}$ are orthogonal bases of $\mathcal{F}_i$ and $\mathcal{G}$ respectively.

The resulting sensitivity indexes proposed by Da Veiga (2015) are defined for each input factor $X_i, i \in \{1, ..., M\}$ as:

$$S_{X_i}^2 = \frac{HSIC(X_i, Y)_{\mathcal{F}_i, \mathcal{G}}}{\sqrt{HSIC(X_i, X_i)_{\mathcal{F}_i, \mathcal{F}_i} HSIC(Y, Y)_{\mathcal{G}, \mathcal{G}}}}. \tag{20}$$

The HSIC measure fully depends on the choice of the RKHS $\mathcal{F}_i$ and $\mathcal{G}$ and especially on the choice of the scalar product that defines the relation between elements from these RKHS. The kernel function defines such scalar product (Meynaoui, 2019). A large range of kernel functions exists and in this paper we focus on universal kernels (i.e. a kernel that is dense in the space of continuous functions with respect to the infinity norm). Indeed, it is necessary to use a universal kernel to characterize the independence of the variables. If the RKHS $\mathcal{F}_i$ and $\mathcal{G}$ are universal, $HSIC(X_i, Y)$ is equal to 0 if and only if $X_i$ and $Y$ are independent (Gretton et al., 2005b; Da Veiga, 2015). Following De Lozzo and Marrel (2014, 2016) and Da Veiga (2015) we chose a Gaussian kernel as they often perform well (Gretton et al., 2005a) and can be used for scalar or vectorial variables. For a vectorial variable $\mathbf{x} \in \mathbb{R}^q$, it is expressed as follows:

$$k(\mathbf{x}, \mathbf{x}') = \exp(-\lambda ||\mathbf{x} - \mathbf{x}'||_2^2), \tag{21}$$





with $||.||_2$ is the Euclidian norm in $\mathbb{R}^q$ and where the hyperparameter $\lambda$ is called the bandwidth parameter of the kernel. In this

study, the bandwith $\lambda$ is estimated from the inverse of the empirical standard deviation of the sample.

**Computation**

The HSIC measure can be rewritten in terms of kernels (Gretton et al., 2005a):

$$HSIC(X_i, Y)_{\mathcal{F}_i, \mathcal{G}} = \mathbb{E}[k_{X_i}(X_i, X_i')k_Y(Y, Y')]$$
$$+ \mathbb{E}[k_{X_i}(X_i, X_i')]\mathbb{E}[k_Y(Y, Y')]$$
$$- 2\mathbb{E}[\mathbb{E}[k_{X_i}(X_i, X_i')|X_i]\mathbb{E}[k_Y(Y, Y')|Y]], \tag{22}$$

where $k_{X_i}$ (resp. $k_Y$) is the kernel function defining the RKHS associated to $X_i$ (resp $Y$), $\underline{\mathbf{X}}' = (X_1', .., X_i', ..., X_M')$ is an

independent and identically distributed copy of $\underline{\mathbf{X}} = (X_1, .., X_i, ..., X_M)$ and where $Y'$ is the output associated to $\underline{\mathbf{X}}'$. Using

Eq. (22), an estimator of HSIC can therefore be computed from an $N$-sample $(x_i^j, y^j), j \in \{1, .., N\}$ of $(X_i, Y)$ such as proposed

in Gretton et al. (2005a):

$$\widehat{HSIC}(X_i, Y)_{\mathcal{F}_i, \mathcal{G}} = \frac{1}{(N-1)^2} Tr(KHLH), \tag{23}$$

where $H \in \mathbb{R}^{N \times N}$ is the centering matrix $H_{ij} = \delta_{ij} - \frac{1}{N}$ and $K \in \mathbb{R}^{N \times N}$ and $L \in \mathbb{R}^{N \times N}$ are the Gram matrices defined as

$K_{ij} = k_{X_i}(x_i, x_j)$ and $L_{ij} = k_Y(y^i, y^j)$.

The HSIC sensitivity indices can be thus easily and quickly computed from a preexisting sample but might be biased if the

sample size is too small as pointed out by De Lozzo and Marrel (2014).

**Ranking and screening and based on HSIC measure**

The HSIC dependence measure can be computed for each input factor $X_i$ and directly used as a sensitivity index for ranking.

In De Lozzo and Marrel (2014) a statistical test approach is also proposed for screening. It is based on bootstraps and it is then

adapted for non asymptotic contexts. Considering an experimental design of $N$ points $(X_i^1, ... X_i^N)$ and the associated output

points $(Y_1, ..., Y_N)$, De Lozzo and Marrel (2014) aim at testing the hypothesis "$X_i$ *and Y are independent*". An estimator

$\widehat{HSIC(X_i}, Y)$ of the dependence measure $HSIC(X_i, Y)$ is firstly computed. Then $B$ bootstrap versions $\underline{\mathbf{Y}}^{[1]}, ..., \underline{\mathbf{Y}}^{[B]}$ are

resampled from the original output sample $(Y_1, ..., Y_N)$ with replacement so as to contain the same number of points $N$. For

each $\underline{\mathbf{Y}}^{[B]}$ the input points associated to $X_i$ are not resampled. Indeed, under the independence hypothesis, any values of Y can

be associated to $X_i$. For each bootstrap version $b$, an estimator $\widehat{HSIC}^{[b]}(X_i, Y)$ is computed. Then, the associated bootstrapped

$p$-value is given by :

$$p\text{-val}_B = \frac{1}{B} \sum_{b=1}^{B} \mathbb{1}_{\widehat{HSIC}^{[b]}(X_i, Y) > \widehat{HSIC}(X_i, Y)} \tag{24}$$

Finally, denoting $\alpha$ the significance level, if $p$-val$_B < \alpha$, the independence hypothesis is rejected, otherwise it is accepted.





### 2.3.3 Random Forest

This section briefly introduces the random forest metamodelling technique and details how it can be used to derive sensitivity measures. Again, it is assumed that the output Y is scalar but the concepts and algorithms below can be extended to multidimensional variables.

**General formulation**

Random forest (Breiman, 2001) belongs to ensemble machine learning techniques. It consists in averaging results from an ensemble of $K$ decision trees created independently. A decision tree is composed of an ensemble of discriminatory binary conditions contained in nodes. Such conditions are hierarchically applied from a root node to a terminal node (tree leaf) (Rodriguez-Galiano et al., 2014). The input space is therefore successively partitioned into smaller groups that correspond to the nodes according to a response variable. Such splitting goes on until reaching a minimum threshold of members per node.

As each individual decision tree is very sensitive to the input dataset, bagging is used to avoid correlations between them and to ensure model stability. It consists in training each decision tree from a different training dataset smaller than the original one. Such subsets are built from the original one by resampling with replacement making some members be used more than once while others may not be used. Such a technique makes the random forests more robust when facing slight variations in the input space and increases accuracy of the prediction (Breiman, 1996, 2001). The samples that are not used to grow a tree are called "out-of-bag" (OOB) data and will be used for the test step. RF workflow is summarized in Figure 4.

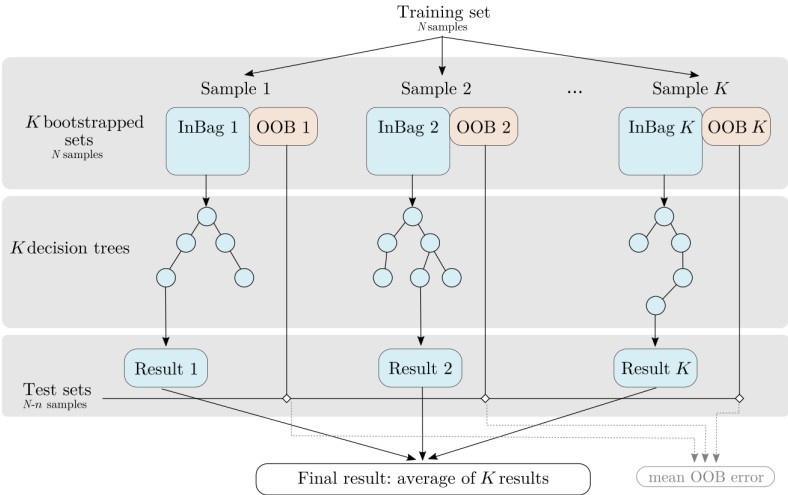

**Figure 4.** RF workflow (adapted from Rodriguez-Galiano et al. 2014). $K$ bootstrapped sets are extracted from the original training set. Part of each set "InBag" is used to grow an independent decision tree and the final regression value is the average of all trees. The remaining portion of each tree "out-of-bag" (OOB) is used as a test set.





**Feature importance for the sensitivity analysis**

RF structure can be cleverly used to provide knowledge about how influential each input factor is. This measure is referred to as *feature importance* in the RF formalism. The random forest is first trained on the targeted output variable Y using a

$N$-points sample $(\underline{\mathbf{X}}^j, \mathrm{Y}^j)$ for $j \in \{1, ..., N\}$. Once the forest has been trained, each input factor $\mathrm{X}_i$ is permuted individually so as to break the link between $\mathrm{X}_i$ and Y. The effect of such permutation on the model accuracy is then investigated. A large decrease in accuracy indicates that the input factor is highly influential whereas a small decrease in accuracy indicates that it has little influence. Different algorithms exist to compute such Mean Decrease in Accuracy (MDA) (see Bénard et al. (2021) for an extensive review of the different formulations in the existing R and Python packages) and we focus here on the original

formulation from Breiman paper (Breiman, 2001). The decrease in accuracy is originally computed from the mean square error between predictions from OOB data with and without permutation for each tree. Results are then averaged over all trees to get the MDA. The algorithm for feature importance calculation is extensively described in various papers (Soleimani, 2021; Bénard et al., 2021, e.g.) and it is reminded in what follows:

1. For each tree $k$:

– Estimate $\hat{\epsilon}_{OOB_k}$ the error from the OOB sample $\mathcal{L}_k$:

$$\hat{\epsilon}_{OOB_k} = \frac{1}{|\mathcal{L}_k|} \sum_{j \setminus (\underline{\mathbf{X}}^j, \mathrm{Y}^j) \in \mathcal{L}_k} (\mathrm{Y}^j - \hat{\mathcal{M}}_{RF}(\underline{\mathbf{X}}^j))^2. \tag{25}$$

Where $\hat{\mathcal{M}}_{RF}$ is the estimated RF metamodel.

– For each input factor $\mathrm{X}_i$:

– Randomly permute $\mathrm{X}_i$ in $\{\underline{\mathbf{X}}^j \in \mathcal{L}_k\}$ to generate a new input set $\{\underline{\mathbf{X}}^{j*} \in \mathcal{L}_k\}$.

– Estimate $\hat{\epsilon}^*_{OOB_k}(i)$ using the permuted input set:

$$\hat{\epsilon}^*_{OOB_k}(i) = \frac{1}{|\mathcal{L}_k|} \sum_{j \setminus (\underline{\mathbf{X}}^j, Y^j) \in \mathcal{L}_k} (\mathrm{Y}^j - \hat{\mathcal{M}}_{RF}(\underline{\mathbf{X}}^{j*}))^2. \tag{26}$$

2. For each input factor $X_i$

– Compute the mean decrease in accuracy $MDA_i$:

$$MDA_i = \frac{1}{K} \sum_{k=1}^{K} \hat{\epsilon}_{OOB_k} - \hat{\epsilon}^*_{OOB_k}(i). \tag{27}$$

Where $K$ is the total number of trees

Despite the "black-box" aspect of RF building, recent works theoretically established a link between Mean Decrease in Accuracy and Sobol Total Indices when input parameters are assumed independent. Indeed, Gregorutti et al. (2017) established that for all input parameter $\mathrm{X}_i$:

$$ST_i = \frac{MDA_i}{2\mathrm{Var}[\mathrm{Y}]}. \tag{28}$$





## 2.4 GSA strategy for the landscape analysis

In this study, several GSA approaches that define the notion of sensitivity in contrasted ways were tested and compared. These methods are described from a methodological point of view in the previous sections and qualitatively summarized in what follows.

A variance decomposition method was first used and the associated Sobol' indices (Sobol, 1993) were computed. These indices measure the impact of each input parameter on the variance of the output. They capture the direct impact of any input and also accurately describe the impact of input parameters when they are interacting ones with others. Sobol' indices direct computation requires a large sample size that could not be afforded in this case study. As a result, we computed Sobol' indices from a limited sample size, based on Polynomial Chaos Expansion (PCE, Sobol 1993; Sudret 2008) in order to circumvect such difficulty. This approach consists in building a surrogate model which analytical polynomial expression is directly related to Sobol' indices. Building a PCE and deducing the associated Sobol' indices thus only requires a training sample of limited size and knowledge about each input parameter probability distribution. In this study, we used the UQLab Matlab software (Marelli and Sudret, 2014) and we computed Sobol' indices from a 1,000-point Latin Hypercube Sample (LHS, McKay et al. 1979). We used a q-norm- and degree-adaptive sparse PCE based on Least Angle Regression Scheme (LARS, Blatman and Sudret 2011) with q-norm $\in [0.1, 0.2, ..., 1.0]$ and a maximum degree of 3.

Second, we computed sensitivity measures based on Hilbert-Schmidt Independence Criterion (HSIC, Da Veiga 2015). HSIC sensitivity indices belong to the category of dependence measures that quantify, from a probabilistic point of view, the dependence between each input and the output. More precisely, the HSIC indice calculates the distance between the input/output joint pdf and the product of their marginal pdfs when they are embedded in a Reproducing Kernel Hilbert Space. HSIC-based measures were used both for ranking and for screening purposes as they allow for fully characterizing the independence between two variables. In addition, these indices can be estimated from small samples (a few hundred of points) and do not depend on the number of inputs. In this study, screening was performed on a 4,000-point LHS to ensure a reasonable simulation time using a statistical test for independence. The power of the test $\alpha$ was set to 1% and 100 bootstrap replicates were used. Then, ranking was performed on the new 1,000-point LHS obtained from the reduced set of input parameters. The R code provided in De Lozzo and Marrel (2016) to compute HSIC measure was adapted in Python to perform both screening and ranking.

Finally, sensitivity indices deduced from Random Forest (RF, Breiman 2001) surrogate model were also computed. A Random Forest model is an empirical input/output relationship based on an ensemble of decision trees. The relative importance of each input parameter in the RF building (called *feature importance measure*) can be easily deduced from it. More precisely, an input parameter $X_i$ is considered important if when breaking the link between $X_i$ and the output Y by permutation, the RF prediction error increases. The metric for measuring prediction error can be defined in different ways and we chose the original formulation of Mean Decrease Accuracy from Breiman (2001). These feature importance measures constitute a kind of sensitivity indices that can be calculated (almost) for free as soon as the RF structure is available. In this study, we used the randomForestSRC R package (Ishwaran and Kogalur, 2020) to obtain feature importance measures once again from the 1,000-point LHS. The number of trees used to train the RF was set to 500.





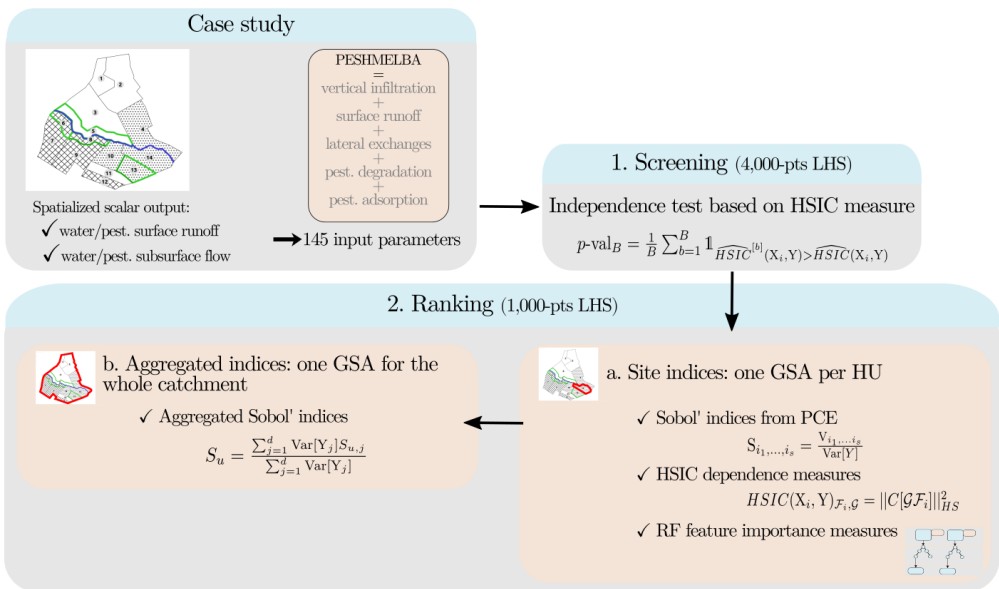

**Figure 5.** Full workflow used to perform GSA on the PESHMELBA model.

In order to investigate PESHMELBA abilities to properly represent transfers in a heterogeneous landscape, sensitivity anal-
405   ysis was performed on spatialized hydrological and quality variables including cumulated water volume and pesticide mass
transferred in the subsurface (saturated lateral transfers) and on surface (surface runoff). The full workflow is summarized in
Figure 5. GSA was first performed at a site scale using each landscape element individually and considering Sobol' indices,
HSIC dependence measures and RF feature importance measures. 95% confidence intervals were also calculated for all meth-
ods and an additional 200-point test set allowed for assessing PCE and RF metamodel performances. The bootstrap resampling
410   procedure provided in UQLab was used to calculate confidence intervals on PCE-based Sobol indices (see Marelli and Sudret
2018 for justification). As RF building already includes a bootstrap step to build each tree, applying another bootstrap could
thus affect the consistency of the OOB sample. We then used a subsampling approach without replacement with subsample size
set to 800 to get error bounds on feature importance measures. As deducing the RF structure from a reduced dataset may lead
to limited performances, we primarily checked that RF performances were reasonably decreased when shifting the training
dataset size from 1,000 to 800. The same procedure was applied to estimate error bounds on HSIC indices as 800 points was
supposed to be far enough to compute consistent estimators. 100 replications were used for all resampling methods.

In a second time, aggregated sensitivity indices were computed to summarize all *site*/local sensitivity indices. In that case, the
objective function was no longer a scalar but rather a vector that gathered a given cumulated variable for all HUs over the
catchment. Aggregated Sobol' indices were calculated following Gamboa et al. (2013) on generalized Sobol' indices (Section
420   2.3.1).





## 2.5 Input sampling

Definition of uncertainty in input factors is critical to make sure that GSA results are representative of the model performance (Sobol, 2001; Muñoz-Carpena et al., 2010; Nossent and Bauwens, 2012; Herman et al., 2013). Such an uncertainty is represented by attaching to each input factor a probability density function (pdf) that reflects its variability in the simulation context

(Gatel et al., 2019). GSA performed in this paper was fully oriented toward the application case described in the previous sections. Therefore, the input factor distributions were chosen as being as representative as possible of the available data on the catchment and the associated uncertainties. Mean values were taken from the standard scenario described in Section 2.2. Distributions and standard deviations were assigned based on experimental measurements from the catchment of application, available scientific literature or expert knowledge (Table 2).

As commonly found in the literature (Coutadeur et al., 2002; Fox et al., 2010; Schwen et al., 2011; Dairon, 2015; Dubus et al., 2003; Dubus and Brown, 2002), lognormal distribution was assigned to saturated hydraulic conductivity. A 20 % coefficient of variation (CV) was used so as to remain representative of each soil horizon hydrodynamical behavior. Distributions for Schaap-Van Genuchten parameters could not be found in the literature, thus $L$ was assigned a uniform distribution +/- 20% around the mean value (Zajac, 2010). As $Ko$ has the same physical meaning than $Ks$, a log-normal distribution was also assigned to

this parameter and a 20 % CV was set. $thetas$, $thetar$ and Van Genuchten parameters $hg$ were assigned normal distributions (Schwen et al., 2011; Alletto et al., 2015; Dairon, 2015; Gatel et al., 2019). A 10% CV was set to $thetas$ (Gatel et al., 2019; Lauvernet and Muñoz-Carpena, 2018) and $thetar$ was assigned a 25% CV (Gatel et al., 2019). A 10 % CV was set for $mn$ and $hg$ (Schwen et al., 2011; Alletto et al., 2015; Gatel et al., 2019). A uniform distribution was assigned to $moc$ (Lauvernet and Muñoz-Carpena, 2018).

Bounds from data were used when several measurements were available. A mean distribution spread was calculated from horizons with several data and was used for setting max and min bounds on horizons when a single measurement was available. A triangular distribution was assigned to $Koc$ (Lauvernet and Muñoz-Carpena, 2018) and a normal distribution was assigned to $DT50$. 60% CV were assigned to $Koc$ and $DT50$ distributions as such parameters are considered relatively uncertain (Dubus et al., 2003). Triangular distributions were assigned to Manning's roughness on plots and for the river bed (Lauvernet and

Muñoz-Carpena, 2018; Gatel et al., 2019). A uniform distribution with a +/- 20 % range around the mean value was assigned to remaining input factors as little information could be found in the literature (Zajac, 2010).

Using a fully distributed model such as PESHMELBA raised the issue of sampling strategy. Indeed, in this case study, even if the site was only composed of 14 surface units, the large number of soil horizons on the catchment, considering the hydrodynamical distinction between plots and VFSs, also dramatically increased the number of parameters. Sampling all

parameters on each spatial unit led to a huge number of simulations that could not be computationally afforded. Moreover, such independent sampling on a very large number of parameters may lead to misinterpretation of the sensitivity analysis results as the influence of physical processes could not be distinguished from spatial arrangement. For each sample, one set of hydrodynamical parameters was therefore sampled for each soil horizon and those parameters were applied to all spatial units that contained this horizon leading to a total of 145 parameters to be sampled.





| Input factor [units] | DescriptioN(Pdf | |
|---|---|---|
| **Soil parameters** | | |
| $thetas$ [m³m⁻³] | Saturated water content | N |
| $thetar$ [m³m⁻³] | Residual water content | TN |
| $Ks$ [ms⁻¹] | Saturated hydraulic conductivity | LN |
| $hg$ [m] | Air-entry pressure in Van Genuchten retention characteristic curve | N |
| $mn$ | Deduced parameter from Van Genuchten retention characteristic curve $n$: $mn = n$-1 | N |
| $Ko$ [ms⁻¹] | Matching point at saturation in modified Mualem Van Genuchten conductivity curve (Schaap and van Genuchten, 2006) | LN |
| $L$ | Empirical pore-connectivity parameter | U |
| $bd$ [gcm⁻³] | Bulk density | U |
| $moc$ [gg⁻¹] | Organic carbon content | U |
| **Pesticide parameters** | | |
| $Kfoc$ [mLg⁻¹] | Freundlich sorption coefficient | T |
| $DT50$ [d] | Half life | N |
| **Vegetation parameters** | | |
| $manning$ [sm⁻¹ᐟ³] | Manning's roughness | T |
| $Zr$ [m] | Rooting depth | U |
| $F10$ | Fraction of the root length density in the top 10% of the root zone | U |
| $LAImax$ | Max LAI value | U |
| **River parameters** | | |
| $hpond$ [m] | Ponding height in the river bed | U |
| $di$ [m] | Distance between the river bed and the limit of impervious saturated zone | U |
| $Ks$ [ms⁻¹] | Saturated conductivity of the river bed | U |
| $manning$ [sm⁻¹ᐟ³] | River bed Manning's roughness | T |
| **Plot and VFS parameters** | | |
| $hpond$ [m] | Ponding height | U |
| $adsorpthick$ [m] | Mixing layer thickness | U |

**Table 2.** Input factor description and corresponding pdf. U:uniform, N:normal, TN:truncated normal, LN:log-normal, T:triangular. Distribution parameters (mean, standard deviation, bounds) are not indicated in this table as they differ from one soil type to another and from one vegetation type to another. Statistics of each assigned distribution are given in detail in Appendix B.





# 3 Results

## 3.1 Screening

Screening was performed at a site scale, on each HU individually, to remain as conservative as possible. Influential parameters at the catchment scale were then deduced from the union of influential parameters for each site. After screening and union, 42 influential parameters are selected for water subsurface flow, 54 parameters are selected for pesticide subsurface flow, 43 parameters are selected for water surface runoff while 45 parameters are retained for pesticide surface runoff. The remaining parameters are given in Appendix C1. The number of input parameters retained after screening remains quite high proving that performing screening on PESHMELBA variables is a challenging task.

It is commonly stated than only a few parameters explain most of the variability but such conclusion does not apply in this case as more than 40 parameters are identified as influential for each output variable considered (see Figure 6). This can be partially explained by the methodology that may not be discriminating enough but it can also be a consequence of the many physical processes interacting in PESHMELBA in a spatially-distributed way, each of them with its own set of characteristic parameters. Additionally, spatial heterogeneities in terms of number of influential parameters are noticed depending on the studied output state variable, as shown in Figure 6. More influential parameters are retained on the right bank (bottom part of the catchment), which is mainly composed of HUs from soil 2 and soil 3 (see Figure 2) for all output variables. Additionally, screening on pesticide variables results on a higher number of influential parameters on HUs situated on the right bank, close to the outlet. Once again, such spatial heterogeneities in the influential parameters identified may be related to the physical processes at stake in the different parts of the catchment. To be noted that no strong correlations are noticed between the locations where pesticides were applied and the number of screened parameters (see Figure 1). Such heterogeneities may rather be related to the climate forcing chosen for the simulation. Additional studies in contrasted climate contexts should be carried out to check whether or not these conclusions can be generalized.

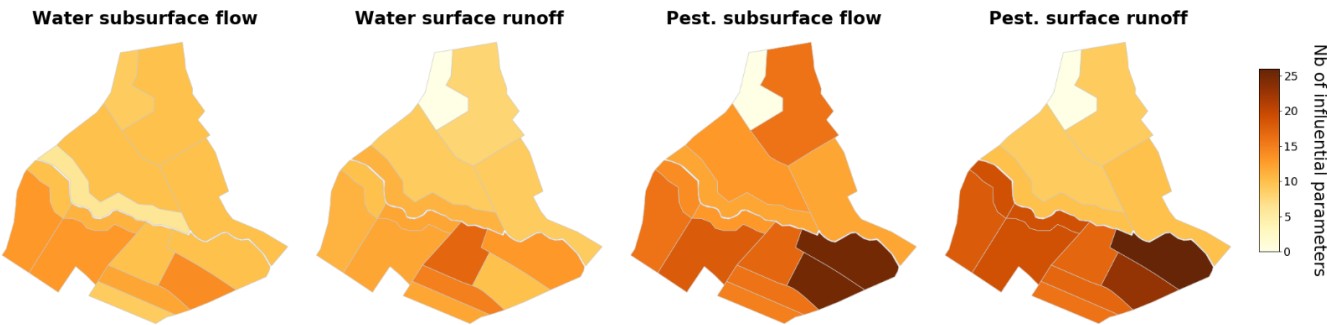

**Figure 6.** Number of influential parameters retained after screening for the different state variables on each HU. From left to right: cumulated water lateral transfers in subsurface, cumulated lateral pesticide transfers in subsurface, cumulated water surface runoff, cumulated pesticide surface runoff.





## 3.2 GSA on a single HU

The three methods (Sobol' indices from PCE, HSIC and RF) were applied on each HU based on the 1,000-point sample generated from the set of screened input parameters. First column of Figure 7 shows the top parameters ranked from Sobol' total-order indices for all output variables on one HU close to the outlet, HU14. The identified influential parameters highly depend on the output variable considered. They are linked to distinct physical processes that may interact with each others and they bring knowledge about the way PESHMELBA represents the hydrological functioning of the virtual catchment. Water subsurface flow (top line) is driven by deep soil hydrodynamical parameters both related to vertical infiltration and subsurface saturated transfers. Water surface runoff (line 2) is also mainly influenced by deep soil parameters. Overland flow is therefore identified as being mostly due to saturation rather than to rainfall excess. Subsurface exchanges with the river is also identified as an influential process as the river bed saturated conductivity ($Ks\_river$) is part of the most influential parameters. Such finding is consistent with the position of HU14, which is directly connected to the river but also to many plots (see Figure 1). Pesticide variables (line 3 and 4) are influenced by a higher diversity of parameters that characterize contrasted and interactive physical processes. Both pesticide subsurface flow and surface runoff are mostly influenced by the pesticide adsorption coefficient. Moreover, unlike the hydrological variables, parameters for surface and intermediary soil horizons rank among the top influential parameters. Such parameters may be linked to vertical infiltration but also to pesticide adsorption, as organic carbon content ($moc$) and saturated water content ($thetas$) are involved in the calculation of adsorption equilibrium. Additionally, the roughness coefficient ($manning\_plot$) and the ponding height ($hpond\_plot$) that are related to surface runoff calculation are also ranked as highly influential on pesticide surface runoff.

Sobol' first-order indices (Figure 7, column 2) reveal that first-order effects explain more than 95% of the water subsurface flow variance and that interactions (defined as $S_T - S_i$) contribute relatively little to the output variance for this variable. The conclusions for remaining variables are much more contrasted as interactions explain more than 40% of the output variance for water surface runoff, more than 70% of pesticide subsurface flow variance and reach more than 80% of explained variance in the case of pesticide surface runoff. Such strong interactions are not only linked to the numerous parameters used to simulate a given physical process in PESHMELBA but also reflect interactions between physical processes in the model.

Columns 3 and 4 in Figure 7 show HSIC and RF sensitivity indices for Sobol' top-ten parameters. On the whole, the rankings obtained from Sobol' total-order, HSIC and RF sensitivity indices are consistent giving confidence in the robustness of these methods. The most influential parameters identified from Sobol' indices are also captured by the other methods. Additional results (not shown here) also show that the top-ten rankings based on Sobol' total indices at least contain the five most influential parameters based on HSIC and RF rankings. Then, Figure 7 does not miss any preponderant parameters for HSIC and RF. Rankings match perfectly well for water subsurface flow while there are slight differences for water surface runoff. Differences between rankings are more pronounced for pesticide variables. In that case, most parameters have zero or very low first order Sobol' indices characterizing nearly purely interacting effects.





**Figure 7.** Sobol' total and first order indices computed using PCE, HSIC and RF site sensitivity indices for all output variables on HU14 with associated 95% confidence intervals. RF feature importance measures are normalized by $2\mathrm{Var}[Y]$ following Eq. (28). HU14 is displayed with a red contour on top left figure. For all methods, displayed parameters are the 10 most influential parameters regarding Sobol' total indices. The bar colours are related to physical processes: brown is related to soil parameters and the darker the brown, the deeper the parameter, blue is related to river parameters whereas green is related to vegetation parameters. Filling in brown bars refers to the soil type of the parameter: soil 1 is not filled, soil 2 is cross hatched whereas soil 3 is filled with circles.





In De Lozzo and Marrel (2016), the authors call for caution when drawing general conclusion about HSIC and Sobol' indices
comparison. In their paper, HSIC indices were found to be more similar to Sobol' first-order than to total-order indices but our
findings reach the opposite conclusion. These results confirm guidelines on the use of these indices stating that they should
not be used for general comparison, being based on different mathematical concepts. While Sobol' indices scrutinize only the
variance of the output, the HSIC indices consider the whole dependence between an input and the output. In addition, HSIC
indices are not intrinsically built to capture interacting effects. Our results show that they obviously capture some interactions
but not all of them and not always.

RF feature importance indices have been proven to relate to Sobol total indices (Gregorutti et al., 2017) but our results exhibit
large discrepancies between RF feature importance measures and Sobol total indices. Table from Appendix D1 provides qual-
ity scores for both PCE and RF metamodels. Results show that the RF metamodel performs much more poorly than the PCE
for all variables probably due to the limited sample size. Despite this limited performances, error bounds on RF indices are
quite small contrarily to error bounds on Sobol' indices estimates. We incriminate the resampling strategy that differ between
the two methods. While the bootstrap technique has been proven to assess the quality of the PCE (Marelli and Sudret, 2018),
the subsampling technique set on RF only targets the precision of feature importance measures. A more adapted subsampling
approach, for example based on Ishwaran and Lu (2019) should probably be further investigated on bigger samples to better
compute RF sensitivity indices and to accurately assess their quality.


To conclude, Sobol' variance decomposition is the only method that gives insight into main and interactive effects bringing
precious knowledge for model validation. Sobol' indices then seem more appropriate to analyse complexe pesticide-related
variables, dominated by physical interactions. These indices also have the merit of being easy to interpret. In addition, a
generalized formulation for aggregated indices is provided by Gamboa et al. (2013) and has already been deeply explored.
However, the precision of the Sobol' indices obtained from PCE remains quite low and results should be thus interpreted with
caution. Considering a high dimension problem and a very limited computational budget (inferior to $1,000$ simulations), HSIC
indices may not be discarded to compute accurate sensitivity indices as soon as detailed knowledge about interactive effects is
not needed.

### 3.3   Landscape analysis

The previous section showed that Sobol' indices (first and total order) provide valuable information about interactions between
parameters. As a result, we focus on this method in what follows. Site rankings such as presented in Figure 7 are gathered
for all HUs in the form of sensitivity maps in Figure 8 for water surface runoff, and in Figure 9 for pesticide surface runoff.
Broadly speaking, both maps show strong spatial heterogeneities regarding influential parameters and a contrasted behaviour
between right and left banks can be identified. For both output variables, hydrodynamical parameters ($thetas$, $thetar$ and $mn$)
of deep horizon from soil 1 (resp. 2 and 3) are mainly influential only on HUs characterized by soil 1 (resp. 2 and 3) (see Figure
2 for a reminder on soil types).

Local hydrodynamical parameters are found to be dominating to explain the output variable variance. A particular case is HU4

(indicated by an array on both Figure 8 and 9) which is characterized by soil 3 while parameters from soil 1 explained most of the variance of both water and pesticide surface runoff variables. The location of HU4, near the outlet, downstream several

soil-1 HUs, may explain such spatial interactions. In addition to specific soil parameters, other parameters such as the manning roughness on vineyard plots ($manning\_plot$) or the coefficient of adsorption ($Koc\_pest$) have a greater influence on HUs from the left bank (top part of the catchment in the Figure).

Finally, comparison of first-order and total-order maps shows quite similar results for water surface runoff on the one hand. It indicates that direct effects are significant for all influential parameters. On the other hand, direct effects are far from dominant

on pesticide surface runoff. Once again, most parameters are influential nearly only in interaction with other parameters since the fist order indices are very low compared to the total order indices.

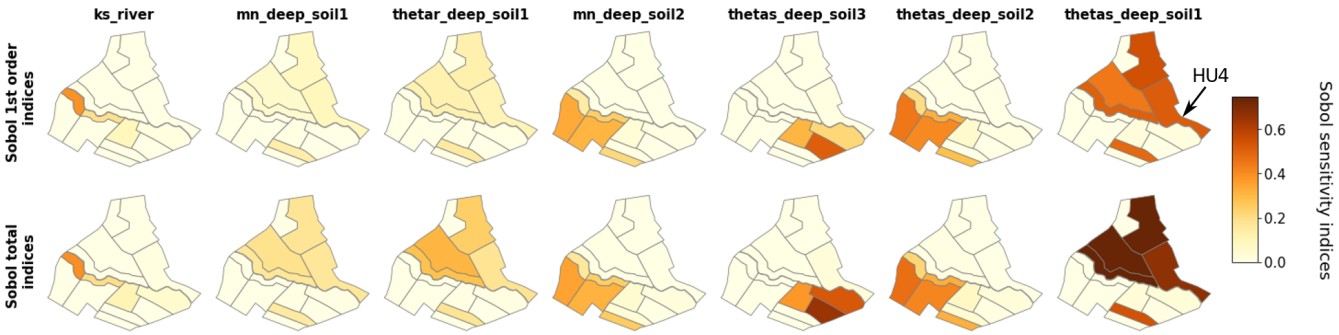

**Figure 8.** Maps of Sobol site sensitivity indices for water surface runoff for the most significantly influential parameters.

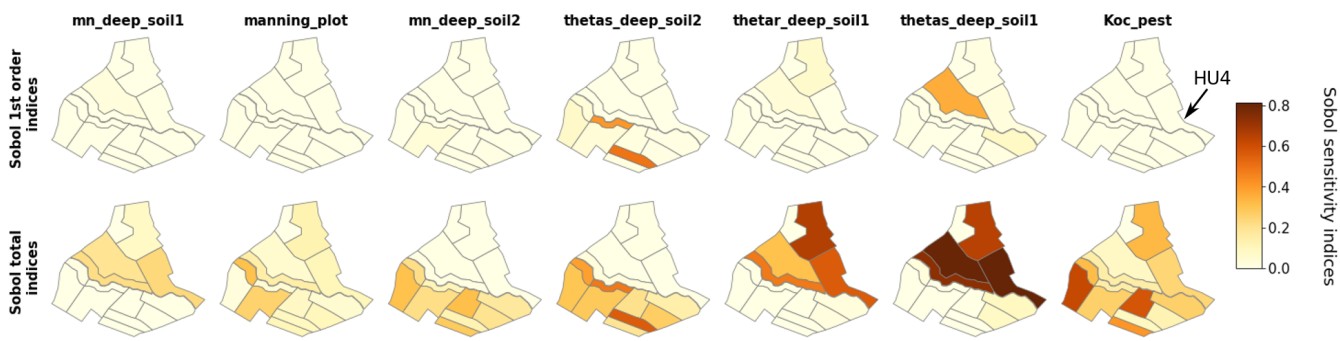

**Figure 9.** Maps of Sobol site sensitivity indices for pesticide surface runoff for the most significantly influential parameters.

Finally, Figure 10 shows aggregated sensitivity indices for water and pesticide surface runoff variables following Gamboa et al. (2013). Since the two banks have contrasted behaviors, aggregated indices are first calculated at the intermediary scale of the bank, then at the catchment scale. For both output variables, rankings strongly differ on each slope. As proposed at a local

scale in Section 3.2, aggregated indices at this scale may constitute a summarized information about the physical processes dominating in PESHMELBA to explain the output variable. For water surface runoff, hydrodynamical soil parameters related





to vertical infiltration ($thetas$, $thetar$ and $mn$) dominate. The influence of $ks\_river$ is only significant in the right bank. The difference of altitude between right and left bank may explain this contrast in the activation of saturated exchanges between water tables and the river. For pesticide surface runoff, deep horizon parameters from soil 1 and pesticide adsorption coefficient

($Koc$) explain a major portion of the output variance on the left bank while pesticide half-life time ($DT50$) and surface runoff parameter ($hpond\_plot$) have lower or no impact. On the contrary, surface parameters ($manning\_plot$ and $hpond\_plot$) have a higher impact on the right slope. In that bank, soil horizons are characterized by lower permeabilities that may result in stronger surface runoff generation than on left bank. In addition pesticide parameters ($Koc$ and $DT50$) are also more influential. More broadly, these results show that pesticide surface runoff may result from the activation and interactions of more physical

processes on the right bank than on left bank.

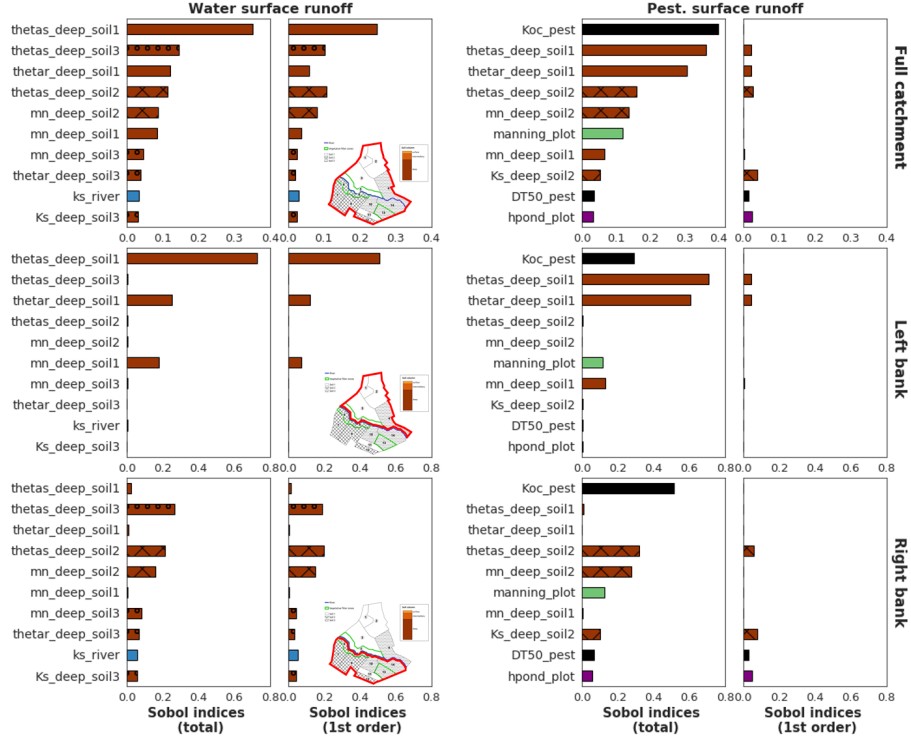

**Figure 10.** Sobol' first-order and total-order aggregated sensitivity indices for water surface runoff (left) and pesticide surface runoff (right) calculated at the scale of the catchment (top), left bank (middle) and right bank (bottom). Displayed parameters are the 11 most influential parameters regarding Sobol indices at the catchment scale for each output variable. The bar colours are related to physical processes: brown is related to soil parameters and the darker the brown, the deeper the parameter is, blue is related to river parameters and green is related to vegetation parameters. Filling in brown bars refers to the soil type of the parameter: soil 1 is not filled, soil 2 is cross hatched while soil 3 is filled with circles.

Sensitivity maps provide local, detailed information about influential parameters on each location of the catchment. However they are computationally costly as one GSA per HU must be performed. This approach may be hard or even impossible to





transpose to real catchment scale composed of several hundreds elements. Catchment scale aggregated indices thus provide a synthetic information at a lower computational price. In this study, Sobol' aggregated indices were directly computed from

site sensitivity indices as they were available. However, if the size of the problem does not allow for direct computation, a pick-and-freeze estimator (Gamboa et al., 2013; De Lozzo and Marrel, 2016) can be used for Sobol' generalized indices. In our case, such overview of sensitivity analysis allows us to focus calibration efforts on deep soil hydrodynamical parameters and pesticide adsorption coefficient to improve the quality of the simulation of both water and pesticide surface flows. As pointed out in Marrel et al. (2015), both approaches are complementary and provide precious knowledge about the model

functioning. The scale of the bank, or more generally of the hillslope, may also constitute an adapted intermediary scale to meet both requirements of detailed results for physical interpretation and computational efficiency.

## 4 Conclusion

In this paper, we have described the first global sensitivity analysis of the modular and coupled PESHMELBA model. For this first experiment, a virtual, simplified catchment was set to explore different approaches for GSA and to propose a methodology

for future real applications. Even if the scenario was simplified, a particular attention was paid to reproduce and to deal with the challenges that would be faced in real applications: high number of input parameters with spatial heterogeneities, limited size of samples due to the high computational cost of a simulation and spatialized ouputs. We first proposed a screening step using an independence test based on the HSIC dependence measure. Then, we compared several methods to compute sensitivity measures: Sobol' indices computed from a PCE surrogate model, HSIC dependence measures and feature importance

measures got from RF surrogate model. Although the results obtained from the three methods were mainly consistent, we noticed that Sobol' analysis was the only method that was able to capture some interactions between parameters, which may be of particular interest in such coupled models.

All methods were first applied on each landscape element individually. They provided us with local measures of sensitivity called site sensitivity indices. Such site sensitivity indices can be gathered into sensitivity maps and they highlight local con-

tributions of parameters. Although very informative about the hydrodynamical functioning of the scenario, these maps were computationally demanding to produce. Then, we used an extension of the previous methods to multidimensional outputs in order to get an overview of sensitivity over the whole physical domain. These aggregated indices were first computed at the catchment scale to characterize the whole output uncertainty. They may allow the users to focus calibration efforts. Additionally, they were computed on each *bank* hillslope and we propose to use this scale as an intermediary scale to get an aggregated

information about the catchment functioning as it still reflects spatial heterogeneities of hydrodynamical processes. When extending this methodology to other case studies, the intermediary spatial scales to be focused on should be defined depending on the characteristics and the goals of the study to make the most of the analysis.

Further research could extend global sensitivity analysis of PESHMELBA to contrasted climatic scenarios in order to encompass different environmental conditions as sensitivity analysis results highly depend on climatic and site condition (Alves Fer-

reira et al., 1995; Lauvernet and Muñoz-Carpena, 2018; Saltelli et al., 2019). Additionally, parameters were assumed to be in-





dependent in this study but this assumption may be too simplistic, especially for hydrodynamical parameters. Further research should adapt the tested methods to dependent parameters. Dealing with dependent parameters has already been explored in the case of Sobol' indices (Chastaing et al., 2015) but it needs to be further explored in the case of HSIC and RF-based methods. It would also be necessary to investigate sensitivity of some time series to get a more comprehensive vision of the model func-

tioning. To do so, the temporal series can be analyzed as a multivariate output for example with clustering-based GSA (Roux et al., 2021) or using the principal components of the model's functional outputs. The definition and the use of adequate hydrological signatures such as proposed in Branger and McMillan (2020) and Horner (2020) may also be of interest to understand space-time variability and to capture a broader range of physical processes.

        Global sensitivity analysis is a necessary but not yet systematic step to model evaluation, especially in the case of spa-

tialized, risk assessment models that can be complex to deal with. This study proposes a comprehensive method based on complementary indices and thus paves the way for systematic analysis of such environmental exposure models.



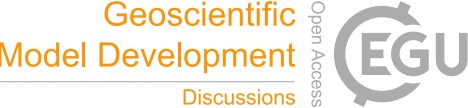

## Appendix

## A Parameters for LAI evolution law

| Parameter | Value | Date |
|-----------|-------|------|
| Vineyard (plot) | | |
| $LAImin$ [-] | 0.01 | February, 1st |
| $LAImax$ [-] | 2.5 | May, 1st |
| $LAIharv$ [-] | 0.01 | November, 10th |
| Grassland (VFS) | | |
| $LAI$ [-] | 5 | - |

**Table A1.** Top: parameters and associated dates used to describe LAI evolution for vineyard cover. Bottom: constant LAI value set on grassland cover.



## B  Input parameter distributions

| | Soil parameters | |
|---|---|---|
| *soilhorizon_thetas_2* | [m³m⁻³] | N(0.3362, 0.00336) |
| *soilhorizon_thetar_2* | [m³m⁻³] | TN(0.0510, 0.0128, 0, 1) |
| *soilhorizon_Ks_2* | [ms⁻¹] | LN(123.04, 7.1) |
| *soilhorizon_hg_2* | [m] | N(-0.0329,0.00329) |
| *soilhorizon_mn_2* | [-] | N(0.1988, 0.0199) |
| *soilhorizon_Ko_2* | [ms⁻¹] | LN(-90.7416,7.128) |
| *soilhorizon_L_2* | [-] | U(-7.8216, -5.2144) |
| *soilhorizon_bd_2* | [gcm⁻³] | U(1.1768, 1.7652) |
| *soilhorizon_moc_2* | [gg⁻¹] | U(0.0024, 0.0054) |
| *soilhorizon_thetas_3* | [m³m⁻³] | N(0.3202, 0.0320) |
| *soilhorizon_thetar_3* | [m³m⁻³] | TN(0.0812, 0.0203, 0, 1) |
| *soilhorizon_Ks_3* | [ms⁻¹] | LN(106.0272, 7.1280) |
| *soilhorizon_hg_3* | [m] | N(-0.0209, 0.00209) |
| *soilhorizon_mn_3* | [-] | N(0.2046, 0.0205) |
| *soilhorizon_Ko_3* | [ms⁻¹] | LN(-47.9772,7.1280) |
| *soilhorizon_L_3* | [-] | U(-5.0844, -3.3896) |
| *soilhorizon_bd_3* | [gcm⁻³] | U(1.2536, 1.8804) |
| *soilhorizon_moc_3* | [gg⁻¹] | U(0.0006, 0.0014) |
| *soilhorizon_thetas_4* | [m³m⁻³] | N(0.2844, 0.0284) |
| *soilhorizon_thetar_4* | [m³m⁻³] | TN( 0.0661, 0.0165, 0, 1) |
| *soilhorizon_Ks_4* | [ms⁻¹] | LN(86.27,7.128) |
| *soilhorizon_hg_4* | [m] | N(-0.0599,0.00599) |
| *soilhorizon_mn_4* | [-] | N(0.2274, 0.0227) |
| *soilhorizon_Ko_4* | [ms⁻¹] | LN(-23.562,7.128) |
| *soilhorizon_L_4* | [-] | U(-0.1716, -0.1144) |
| *soilhorizon_bd_4* | [gcm⁻³] | U(1.2240, 1.8360) |
| *soilhorizon_moc_4* | [gg⁻¹] | U(4.3840 10⁻⁴,9.6160 10⁻³) |
| *soilhorizon_thetas_6* | [m³m⁻³] | N(0.3537, 0.0354) |
| *soilhorizon_thetar_6* | [m³m⁻³] | TN(0, 0.0093, 0, 1) |
| *soilhorizon_Ks_6* | [ms⁻¹] | LN(73.0512,7.128) |
| *soilhorizon_hg_6* | [m] | N-0.066,0.0066 |
| *soilhorizon_mn_6* | [-] | N(0.1289, 0.0129)) |





| | | |
|---|---|---|
| *soilhorizon_Ko_6* | [ms⁻¹] | LN(-78.5664,7.1280) |
| *soilhorizon_L_6* | [-] | U(7.7240, 19.3100) |
| *soilhorizon_bd_6* | [gcm⁻³] | U(1.2704, 1.9056) |
| *soilhorizon_moc_6* | [gg⁻¹] | U(0.0042, 0.0094) |
| *soilhorizon_thetas_7* | [m³m⁻³] | N(0.3247, 0.0325) |
| *soilhorizon_thetar_7* | [m³m⁻³] | TN(0, 0.0093, 0, 1) |
| *soilhorizon_Ks_7* | [ms⁻¹] | LN(43.9416,7.1280) |
| *soilhorizon_hg_7* | [m] | N(-0.0718,0.00718) |
| *soilhorizon_mn_7* | [-] | N(0.0751, 0.0075) |
| *soilhorizon_Ko_7* | [ms⁻¹] | LN(-101.99,7.128) |
| *soilhorizon_L_7* | [-] | U(-12, -8) |
| *soilhorizon_bd_7* | [gcm⁻³] | U(1.3256, 1.9884) |
| *soilhorizon_moc_7* | [gg⁻¹] | U(0.0019, 0.0051) |
| *soilhorizon_thetas_8* | [m³m⁻³] | N(0.4162, 0.0416) |
| *soilhorizon_thetar_8* | [m³m⁻³] | TN(0, 0.0093, 0, 1) |
| *soilhorizon_Ks_8* | [ms⁻¹] | LN(12.2076,7.1280) |
| *soilhorizon_hg_8* | [m] | N(-0.3018, 0.03018) |
| *soilhorizon_mn_8* | [-] | N(0.10000, 0.0100) |
| *soilhorizon_Ko_8* | [ms⁻¹] | LN(-121.39,7.128) |
| *soilhorizon_L_8* | [-] | U(8, 20) |
| *soilhorizon_bd_8* | [gcm⁻³] | U(1.2304, 1.8456) |
| *soilhorizon_moc_8* | [gg⁻¹] | U(0.0018, 0.0037) |
| *soilhorizon_thetas_9* | [m³m⁻³] | N(0.3322, 0.0332) |
| *soilhorizon_thetar_9* | [m³m⁻³] | TN(0.0770, 0.0192, 0, 1) |
| *soilhorizon_Ks_9* | [ms⁻¹] | LN(85.543,7.128) |
| *soilhorizon_hg_9* | [m] | N(-0.0671,0.00671) |
| *soilhorizon_mn_9* | [-] | N(0.2582, 0.0258) |
| *soilhorizon_Ko_9* | [ms⁻¹] | LN(-76.7376,7.128) |
| *soilhorizon_L_9* | [-] | U(0.3376, 0.8440) |
| *soilhorizon_bd_9* | [gcm⁻³] | U(1.1664, 1.7496) |
| *soilhorizon_moc_9* | [gg⁻¹] | U(0.0023, 0.0051) |
| *soilhorizon_thetas_10* | [m³m⁻³] | N(0.3160, 0.0316) |
| *soilhorizon_thetar_10* | [m³m⁻³] | TN(0.0612, 0.0153, 0, 1) |
| *soilhorizon_Ks_10* | [ms⁻¹] | LN(76.565,7.128) |
| *soilhorizon_hg_10* | [m] | N(-0.0356, 0.00356) |



| | | |
|---|---|---|
| *soilhorizon_mn*_10 | [-] | N(0.1791, 0.0179) |
| *soilhorizon_Ko*_10 | [ms$^{-1}$] | LN(-80.827,7.128) |
| *soilhorizon_L*_10 | [-] | U(0.8376, 2.0940) |
| *soilhorizon_bd*_10 | [gcm$^{-3}$] | U(1.2984, 1.9476) |
| *soilhorizon_moc*_10 | [gg$^{-1}$] | U(0.0025, 0.0055)) |
| *soilhorizon_thetas*_11 | [m$^3$m$^{-3}$] | N(0.3375, 0.0338) |
| *soilhorizon_thetar*_11 | [m$^3$m$^{-3}$] | TN(0.0372, 0.0093, 0, 1) |
| *soilhorizon_Ks*_11 | [ms$^{-1}$] | LN(94.6476,7.128) |
| *soilhorizon_hg*_11 | [m] | N(-0.0969,0.00969) |
| *soilhorizon_mn*_11 | [-] | N(0.2685, 0.0268) |
| *soilhorizon_Ko*_11 | [ms$^{-1}$] | LN(-82.5336,7.1280) |
| *soilhorizon_L*_11 | [-] | U(-10.1124, -6.7416) |
| *soilhorizon_bd*_11 | [gcm$^{-3}$] | U(1.0752, 1.6128) |
| *soilhorizon_moc*_11 | [gg$^{-1}$] | U(0.0049, 0.0050) |
| *soilhorizon_thetas*_14 | [m$^3$m$^{-3}$] | N(0.3375, 0.0338) |
| *soilhorizon_thetar*_14 | [m$^3$m$^{-3}$] | TN(0.0372, 0.0093, 0, 1) |
| *soilhorizon_Ks*_14 | [ms$^{-1}$] | LN(96.7824,7.128) |
| *soilhorizon_hg*_14 | [m] | N(-0.0969,0.00969) |
| *soilhorizon_mn*_14 | [-] | N(0.2685, 0.0268) |
| *soilhorizon_Ko*_14 | [ms$^{-1}$] | LN(-82.5336,7.128) |
| *soilhorizon_L*_14 | [-] | U(-10.1124, -6.7416) |
| *soilhorizon_bd*_14 | [gcm$^{-3}$] | U(1.0752, 1.6128) |
| *soilhorizon_moc*_14 | [gg$^{-1}$] | U(0.0175, 0.0385) |
| *soilhorizon_thetas*_12 | [m$^3$m$^{-3}$] | N(0.3375, 0.0338) |
| *soilhorizon_thetar*_12 | [m$^3$m$^{-3}$] | TN(0.0372, 0.0093, 0, 1) |
| *soilhorizon_Ks*_12 | [ms$^{-1}$] | LN(94.6476,7.128) |
| *soilhorizon_hg*_12 | [m] | N(-0.0969,0.00969) |
| *soilhorizon_mn*_12 | [-] | N(0.2685, 0.0268) |
| *soilhorizon_Ko*_12 | [ms$^{-1}$] | LN(-82.5336,7.1280) |
| *soilhorizon_L*_12 | [-] | U(-10.1124, -6.7416) |
| *soilhorizon_bd*_12 | [gcm$^{-3}$] | U(1.0752, 1.6128) |
| *soilhorizon_moc*_12 | [gg$^{-1}$] | U(0.0072, 0.0158) |
| *soilhorizon_thetas*_15 | [m$^3$m$^{-3}$] | N(0.3375, 0.0338) |
| *soilhorizon_thetar*_15 | [m$^3$m$^{-3}$] | TN(0.0372, 0.0093) |
| *soilhorizon_Ks*_15 | [ms$^{-1}$] | LN(96.7824,7.128) |





| | | |
|---|---|---|
| *soilhorizon_hg*_15 | [m] | N(-0.0969,0.00969) |
| *soilhorizon_mn*_15 | [-] | N(0.2685, 0.0268) |
| *soilhorizon_Ko*_15 | [ms$^{-1}$] | LN-82.5336,7.128 |
| *soilhorizon_L*_15 | [-] | U(-10.1124, -6.7416) |
| *soilhorizon_bd*_15 | [gcm$^{-3}$] | U(1.0752, 1.6128) |
| *soilhorizon_moc*_15 | [gg$^{-1}$] | U(0.0175, 0.0385) |
| *soilhorizon_thetas*_13 | [m$^3$m$^{-3}$] | N(0.3375, 0.0338) |
| *soilhorizon_thetar*_13 | [m$^3$m$^{-3}$] | TN(0.0372, 0.0093, 0, 1) |
| *soilhorizon_Ks*_13 | [ms$^{-1}$] | LN(94.6476,7.128) |
| *soilhorizon_hg*_13 | [m] | N(-0.0969,0.00969) |
| *soilhorizon_mn*_13 | [-] | N(0.2685, 0.0268) |
| *soilhorizon_Ko*_13 | [ms$^{-1}$] | LN(-82.5336,7.128) |
| *soilhorizon_L*_13 | [-] | U(-10.1124, -6.7416) |
| *soilhorizon_bd*_13 | [gcm$^{-3}$] | U(1.0752, 1.6128) |
| *soilhorizon_moc*_13 | [gg$^{-1}$] | U(0.0067, 0.0080) |
| *soilhorizon_thetas*_16 | [m$^3$m$^{-3}$] | N(0.3375, 0.0338) |
| *soilhorizon_thetar*_16 | [m$^3$m$^{-3}$] | TN(0.0372, 0.0093, 0, 1) |
| *soilhorizon_Ks*_16 | [ms$^{-1}$] | LN(96.7824,7.128) |
| *soilhorizon_hg*_16 | [m] | N(-0.0969,0.00969) |
| *soilhorizon_mn*_16 | [-] | N(0.2685, 0.0268) |
| *soilhorizon_Ko*_16 | [ms$^{-1}$] | LN(-82.5336,7.128) |
| *soilhorizon_L*_16 | [-] | U(-10.1124, -6.7416) |
| *soilhorizon_bd*_16 | [gcm$^{-3}$] | U(1.0752, 1.6128) |
| *soilhorizon_moc*_16 | [gg$^{-1}$] | U(0.0175, 0.0385) |
| Pesticide parameters | | |
| *pest_Koc* | [mLg$^{-1}$] | T(461.4000, 538.3000, 769.0000) |
| *pest_DT*50 | [d] | N(47.1, 28.26) |
| Vegetation parameters | | |
| *veget_manning*_1 | [sm$^{-1/3}$] | T(0.0250, 0.0330, 0.041) |
| *veget_Zr*_1 | [m] | U(2.096,3.144) |
| *veget_F*10_1 | [-] | U(0.2960, 0.4440) |
| *veget_LAImin*_1 | [-] | U(0.0080, 0.0120) |
| *veget_LAImin*_1 | [-] | U(0.0080, 0.0120) |
| *veget_LAImax*_1 | [-] | U(2, 3) |





| | | |
|---|---|---|
| $veget\_LAIharv\_1$ | [-] | U(0.0080, 0.0120) |
| $veget\_manning\_2$ | [sm$^{-1/3}$] | T(0.1000, 0.2000, 0.3000) |
| $veget\_Zr\_2$ | [m] | U(7.2,1.08) |
| $veget\_F10\_2$ | [-] | U(0.2680, 0.4020) |
| $veget\_LAI\_2$ | [-] | U(4, 6) |
| River parameters | | |
| $river\_hpond$ | [m] | U(0.008,0.012) |
| $river\_di$ | [m] | U(1.2, 1.8) |
| $river\_Ks$ | [ms$^{-1}$] | U( 76.5648, 7.1280) |
| $river\_manning$ | [sm$^{-1/3}$] | T(0.0610, 0.0360, 0.0790) |
| Plot and VFS parameters | | |
| $plot\_hpond$ | [m] | U(0.008, 0.012) |
| $vfz\_hpond$ | [m] | U(0.04, 0.06) |
| $hu\_adsorpthick$ | [m] | U(0.005, 0.015) |

Table B1: Distribution and statistics of the assigned pdfs for the 145 input parameters, uniform:U(min,max), triangular: T(min,mean,max), normal:N(mean,standard deviation), log-normal: LN(mean,standard deviation), truncated normal: TN(mean, standard deviation, min,max).





## C Screening results

| Water subsurface flow | Pesticide subsurface flow | Water surface runoff | Pesticide surface runoff |
|---|---|---|---|
| soilhorizon_thetas_2 | soilhorizon_thetas_12 | soilhorizon_thetas_11 | soilhorizon_thetas_11 |
| soilhorizon_thetas_4 | soilhorizon_thetas_15 | soilhorizon_thetas_15 | soilhorizon_thetas_12 |
| soilhorizon_thetas_6 | soilhorizon_thetas_2 | soilhorizon_thetas_2 | soilhorizon_thetas_13 |
| soilhorizon_thetas_7 | soilhorizon_thetas_4 | soilhorizon_thetas_4 | soilhorizon_thetas_15 |
| soilhorizon_thetas_8 | soilhorizon_thetas_6 | soilhorizon_thetas_6 | soilhorizon_thetas_2 |
| soilhorizon_thetas_10 | soilhorizon_thetas_7 | soilhorizon_thetas_7 | soilhorizon_thetas_4 |
| soilhorizon_thetar_3 | soilhorizon_thetas_8 | soilhorizon_thetas_8 | soilhorizon_thetas_6 |
| soilhorizon_thetar_4 | soilhorizon_thetas_10 | soilhorizon_thetas_10 | soilhorizon_thetas_7 |
| soilhorizon_thetar_8 | soilhorizon_thetar_2 | soilhorizon_thetar_15 | soilhorizon_thetas_8 |
| soilhorizon_thetar_10 | soilhorizon_thetar_4 | soilhorizon_thetar_2 | soilhorizon_thetas_10 |
| soilhorizon_moc_13 | soilhorizon_thetar_8 | soilhorizon_thetar_4 | soilhorizon_thetar_15 |
| soilhorizon_mn_3 | soilhorizon_thetar_10 | soilhorizon_thetar_8 | soilhorizon_thetar_2 |
| soilhorizon_mn_4 | soilhorizon_pore_6 | soilhorizon_thetar_10 | soilhorizon_thetar_4 |
| soilhorizon_mn_6 | soilhorizon_moc_12 | soilhorizon_pore_9 | soilhorizon_thetar_8 |
| soilhorizon_mn_8 | soilhorizon_moc_15 | soilhorizon_mn_11 | soilhorizon_thetar_10 |
| soilhorizon_mn_10 | soilhorizon_moc_2 | soilhorizon_mn_2 | soilhorizon_moc_6 |
| soilhorizon_Kx_3 | soilhorizon_moc_6 | soilhorizon_mn_4 | soilhorizon_moc_12 |
| soilhorizon_Kx_4 | soilhorizon_moc_9 | soilhorizon_mn_6 | soilhorizon_mn_11 |
| soilhorizon_Kx_8 | soilhorizon_mn_11 | soilhorizon_mn_7 | soilhorizon_mn_16 |
| soilhorizon_Kx_10 | soilhorizon_mn_16 | soilhorizon_mn_8 | soilhorizon_mn_2 |
| soilhorizon_Ks_11 | soilhorizon_mn_4 | soilhorizon_mn_10 | soilhorizon_mn_4 |
| soilhorizon_Ks_13 | soilhorizon_mn_6 | soilhorizon_Kx_8 | soilhorizon_mn_6 |
| soilhorizon_Ks_14 | soilhorizon_mn_8 | soilhorizon_Kx_10 | soilhorizon_mn_7 |
| soilhorizon_Ks_15 | soilhorizon_mn_10 | soilhorizon_Ks_12 | soilhorizon_mn_8 |
| soilhorizon_Ks_16 | soilhorizon_Kx_12 | soilhorizon_Ks_13 | soilhorizon_mn_10 |
| soilhorizon_Ks_3 | soilhorizon_Kx_9 | soilhorizon_Ks_15 | soilhorizon_Ks_15 |
| soilhorizon_Ks_4 | soilhorizon_Kx_10 | soilhorizon_Ks_16 | soilhorizon_Ks_2 |
| soilhorizon_Ks_6 | soilhorizon_Ks_12 | soilhorizon_Ks_4 | soilhorizon_Ks_4 |
| soilhorizon_Ks_7 | soilhorizon_Ks_14 | soilhorizon_Ks_6 | soilhorizon_Ks_8 |
| soilhorizon_Ks_8 | soilhorizon_Ks_15 | soilhorizon_Ks_8 | soilhorizon_Ks_9 |
| soilhorizon_Ks_9 | soilhorizon_Ks_16 | soilhorizon_Ks_9 | soilhorizon_hg_3 |
| soilhorizon_Ks_10 | soilhorizon_Ks_2 | soilhorizon_Ks_10 | soilhorizon_hg_4 |




| | | | |
|---|---|---|---|
| *soilhorizon_hg_*16 | *soilhorizon_Ks_*4 | *soilhorizon_hg_*4 | *soilhorizon_hg_*8 |
| *soilhorizon_hg_*2 | *soilhorizon_Ks_*6 | *soilhorizon_hg_*6 | *soilhorizon_bd_*6 |
| *soilhorizon_hg_*4 | *soilhorizon_Ks_*8 | *soilhorizon_hg_*8 | *soilhorizon_bd_*13 |
| *soilhorizon_hg_*8 | *soilhorizon_Ks_*9 | *soilhorizon_hg_*10 | *soilhorizon_bd_*12 |
| *soilhorizon_hg_*9 | *soilhorizon_Ks_*10 | *soilhorizon_bd_*3 | *river_ks* |
| *soilhorizon_bd_*2 | *soilhorizon_hg_*4 | *river_ks* | *river_di* |
| *river_ks* | *soilhorizon_hg_*6 | *river_di* | *plot_hpond* |
| *river_di* | *soilhorizon_hg_*8 | *plot_hpond* | *pest_Koc_*1 |
| *plot_hpond* | *soilhorizon_hg_*10 | *vfz_hpond* | *pest_DT*50_1 |
| *VFS_hpond* | *soilhorizon_bd_*11 | *veget_LAIharv_*1 | *HU_adsorpthick* |
| | *soilhorizon_bd_*12 | *veget_F*10_1 | *veget_Zr_*1 |
| | *soilhorizon_bd_*13 | | *veget_manning_*1 |
| | *soilhorizon_bd_*15 | | *veget_F*10_1 |
| | *soilhorizon_bd_*2 | | |
| | *soilhorizon_bd_*6 | | |
| | *soilhorizon_bd_*9 | | |
| | *river_ks* | | |
| | *river_di* | | |
| | *plot_hpond* | | |
| | *veget_Zr_*1 | | |
| | *pest_Koc_*1 | | |
| | *pest_DT*50_1 | | |

Table C1: Remaining parameters after screening step for each output variable. In the XXX_XXX_XXX syntaxe of parameter names, the first block is the type of element the parameter refers to (soil horizon, river, vegetation, pesticide, HU or VFS), the second part is the parameter name while the last part is the element index the parameter refers to (soil horizon or vegetation type).





# D   Metamodel performances

|  | PCE | RF |
|---|---|---|
| WaterLateralFlow | 0.98 | 0.85 |
| WaterSurfaceRunoff | 0.80 | 0.44 |
| PesticideLateralFlow | 0.75 | 0.55 |
| PesticideSurfaceRunoff | 0.75 | 0.51 |

**Table D1.** $Q^2$ score for all variables on HU14 calculated from the 200-point test set. The $Q^2$ score is calculated as $Q^2 = 1 - \frac{\sum_{i=1}^{N}(\mathcal{M}(\mathbf{X}^i)-Y^i)^2}{\sum_{i=1}^{N}(Y^i-\overline{Y})^2}$, where $\overline{Y} = \frac{1}{N}\sum_{i=1} Y^i$ is the empirical mean of the sample.



*Code and data availability.* The PESHMELBA model is an open-source model coded in Python and Fortran 90 and embedded in the OpenPALM coupler. The code for the OpenPALM coupler is available from www.cerfacs.fr/globc/PALM_WEB/user.html#download af-ter registration. The full PESHMELBA code is freely made available on request from the corresponding author, Emilie Rouzies (emi-
lie.rouzies@inrae.fr). UQLab (www.uqlab.com) is an open-source Matlab software for uncertainty quantification. The Python scripts used to compute HSIC sensitivity indices are freely made available on request from the corresponding author. The Python Scikit-learn package is available at https://scikit-learn.org. All the data produced in this study can also be made available on request from the corresponding author.

*Author contributions.* All authors contributed to writing the text and to all stages of editing. PCE computation was performed by Bruno Sudret and Emilie Rouzies whereas HSIC and RF indices computation was led by Emilie Rouzies with extensive support from Claire
Lauvernet and Arthur Vidard.

*Competing interests.* The authors declare that they have no conflict of interest.

*Acknowledgements.* The authors kindly acknowledge Stefano Marelli for his support on UQlab and PCE usage for GSA and Bertrand Ioss for his advice on the computation on RF feature importance measures.



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
