# Peer review of "How to perform global sensitivity analysis of a catchment-scale, distributed pesticide transfer model? Application to the PESHMELBA model."

_Geoscientific Model Development, 2021_

## Author Response (AR1)

**All the comments and questions of Reviewers 1 and 2 have been copied hereafter in bold. We also provide a revised manuscript as a supplement to these response comments so as a marked-up manuscript version showing the changes made.**

**Reviewer 1**

Thank you very much for the careful review and edits to the initital submission. We have already provided responses to detailed comments during the discussion period. In what follows, we address main comments and duplicate responses to detailed comments. All your comments and questions have been copied hereafter in bold. We have also revised the manuscript accordingly to accommodate them.

**Main comments**

– **The method section focuses too much on the technical details of the sensitivity analysis methods (which are not methods, but methods taken from previous studies ). The methodology (how these methods are used) is not well explained, which I think should be more the focus of the paper.** As suggested, the structure of the paper has been deeply modified so as to get a clearer overview of the full methodology, less technical details about the methods used for sensitivity analysis but more practical considerations on how these methods are used. To do so, Section 2.3 and Section 2.5 have been merged and shortened to keep only the main equations relative to each method. An overview of the full methodology so as a justification for using and comparing several methods has also been added at the begining of Section 2.4

– **Some clarification are needed regarding the setup of the case study (Section 2.2).** According to your detailed comments, the model setup section has been modified so as to provide readers with a clearer description of the parameters considered in this study. The number of parameters envolved in the sensitivity analysis has notably been specifiedearlier in the text. The Table from Section 2.5 also comes earlier and it has been enriched with a description of the different categories of parameters.

– **The manuscript lacks a discussion of the methodology and results with respect to previous studies. This would help to clarify the novelty of the study. In particular:**

**The authors highlight that this is the first sensitivity analysis applied to the PESHMELBA model (e.g. L588 L26), but sensitivity analysis was applied to other pesticide models (e.g. Dubus et al., 2003; Hong & Purucker, 2018...). The manuscript lacks a review on previous sensitivity analyses (local, global) applied to pesticide models.** As suggested, a review on sensitivity analysis applied to pesticide models has been added in the introduction. It covers both the different approaches (local vs. global) and the use of a screening method to decrease the dimension of the problem.

– **It is also not clear to what extent the methodology for sensitivity analysis proposed in the manuscript is new compared to previous sensitivity analysis studies. In this respect, previous studies have also proposed to use first a computationally cheaper sensitivity analysis method (method that requires a relatively low number of model simulations, such as the Morris Elementary EffectTest) to screen non-influential inputs, before applying a computationally more expensive method (e.g. Sobol' Variance Based method) based on the subset of influential inputs (e.g. Garcia et al., 2019; Vanuytrecht et al., 2014). This could be discussed in the manuscript.** Indeed, the methodology we followed to perform sensitivity analysis in this study is a classical approach: first, a screening step and second, a ranking step applied on the reduced set of input parameters. However, for both steps, the specificities af the application (high number of input parameters and high computational cost of a PESHMELBA simulation) prevented us from using classical methods. Alternatives approaches, that are recent and, up to now, poorly applied to pesticide model analysis were then necessary. In addition, combining several ranking methods with different definitions of sensitivity to get a robust overview of influential parameters is also new. A discussion section about the full methodology has been added to argue on these points.

– **I wish to point out that the PESHMELBA model, as well as the code to compute the HSIC sensitivity indices are not publicly available, but are available upon request from the corresponding author (Code and data availability section). To advance open science (and to comply with the GMD guidelines?), I think that it would be valuable to make these resources openly available, especially since the paper has a methodological focus.** In order to comply with GMD Code and Data Policy, two Zenodo repositories have been created to provide both PESHMELBA source code and data. The urls and DOI have been added to the 'Code and Data Availability' section : - PESHMELBA software: https://zenodo.org/record/6319769#.YinMV1TjKUk - Data and codes for sensitivity analysis: https://zenodo.org/record/6319773#.YinMc1TjKUk

**Detailed comments**

– **L21 p1 'simple enough to ensure flexibility': More explanation is needed here. This is vague and I am not sure what is meant by flexibility.** Here we mean that models used to support decision-making should be designed so that users could easily modify the code to integrate new physical processes and/or adapt the existing ones. "Flexibility" then refers to the structure of these tools that should be ideally simple enough to enable such evolutions. The sentence has been clarified in the manuscript.

– **L30-31 p2 'catchment-scale model [...] afforded': Specify that this is spatially distributed models.** Yes, added as suggested.

– **L61-73 p3: Also note the recent study of Smith et al. (2021).** Yes, added as suggested.

– **Section 2.1: a presentation of the model parameters is missing. How many uncertain parameters that needs to be estimated are there? What are the different categories of parameters (e.g. soil, pesticide, vegetation etc., as I can read in Table 2). Parameters are only introduced much later in Section 2.5 (Table 2), which makes it difficult to follow Section 2.2 that describes the selection of the parameter values. The reference to Table 2 in the caption of Table 1 does not flow well.** The model setup section has been deeply modified so as to introduce a presentation of model parameters much earlier than in the original manuscript. The Table from Section 2.5 has been modified and comes earlier to introduce all model parameters and associated categories. A sentence has also been added in the text to specify the number of uncertain parameters envolved in the senstivity analysis.

– **Section 2.2: Why performing the experiment on a virtual catchment and not a 'real' one ?** As mentioned in the text, the final targetted catchment for this study is the real La Morcille catchment. Figure 1 (left) depicts PESHMELBA meshing at this scale showing that such application results in a high number of landscape elements (>500). Conducting experiments at the full catchment scale would have drastically increased the computational cost of the analysis while turning difficult the interpretation of sensitivity analysis results considering that no such experiment has been conducted before. We then perform the experiment on a simplified case as a first try to get a clearer and simpler interpretation of the results both regarding methodological and spatial aspects.

– **I understand that the simulation experiment considers the application of the fungicide at the beginning of the winter period. Is this realistic?** As pointed out, considering an application of fungicide at the begining of the winter period is not very realistic. Actually, we suggest to remove all mention to "winter" period as the focus of this study is mainly methodological, based on a virtual case and realistic forcings. The chosen setup primarily aims at identifying influential factors on different physical processes integrated in PESHMELBA with a strong focus on lateral transfers of water and pesticides. We have then favoured a scenario with strong rain events since they result in both surface runoff and lateral saturated transfers in subsurface. The results of this study then provide general guidelines about the model behaviour but they should be further complemented with applications on each particular agropedoclimatic context of interest.

– **Why performing the experiments over a 3-month winter period? This is a very short time period.** In this case study, PESHMELBA time step is 1h on dry periods and 30 minutes during or after rainfall events resulting in a high computational cost for a three-month simulation (2h per simulation on the cluster used to run the simulations). A longer time

period was then no affordable for this first experiment. In addition, we chose a period characterized by high cumulative rainfall volume to make sure that the different physical processes simulated in PESHMELBA would activate during the simulation (we were mainly concerned with activation of surface runoff and lateral saturated exchanges). This way, the performed sensitivity can also be used as a consistency check on the model structure itself allowing to check different physical processes simulation. However, we remain aware that results from GSA highly depend in climatic conditions as precised in the conclusion of the manuscript. As mentioned, further researches may focus on other contrasted time periods to draw robust conclusions.

– **A justification for the soil moisture initial condition (hydrostatic equilibrium L157) is missing.** An hydrostatic equilibrium has been chosen so as to provide the model with initial conditions as "neutral" as possible. We wanted the variables of interest to fully represent the dynamic of the catchment and not to include any non-physical warm-up period. To do so, another approach consists in running a warm-up simulation on a longer period but it would imply a high computational cost that could not have been afforded in this case.

– **Section 2.3-2.4: I think that section 2.3 provides too many technical details that are not necessary to understand the methodology and analyses presented in the paper. The authors recognize themselves that this section could be skipped L183-184. My suggestion is to report only the main equations used to compute the sensitivity indices, while details on the derivation of these equations (that were taken from previous papers and that are therefore not really a contribution of this paper, if I understand correctly) can be moved in the supplements/appendix. I am mostly referring to the description of the Sobol' and HSIC methods, while I think that the description of the random forest method in Section 2.4 reads very well. The main equations and references of Section 2.3 can be combined with the summary of the GSA methods provided in Section 2.5, to provide the reader only with the information that are needed to understand the methodology and the analyses, while avoiding unnecessary repetitions between Sections 2.4 and 2.5. In addition, I think that an overview of the methodology (why do you need to use the GSA methods?) is needed before introducing the specific GSA methods.** The section on method description has been fully reviewed as suggested. Section 2.3 and 2.5 have been merged and only the main equations relative to each methods now remain together with more practical interpretation of calculated indices. We have also added a justification for method comparison and an overview on the full methodology at the begining of the section.

– **Equation (17): The sensitivity index for a given input is the average of the first order indices estimated for the different model outputs, weighted by the outputs variance, am I correct? This paper aims to help applying these methods, therefore I think that interpreting the equations in simple (intuitive) terms, would improve readability and clarity. It is very nice to have the formal mathematical proof for the equation, but the proof does not have any practical implications and could be moved into the supplements/appendix (this is an example of how this section could be simplified, see my previous comment).** Indeed, aggregated sensitivity indices correspond to an average of Sobol' indices on each landscape unit weighted by local output variances. As suggested, the proof has been removed from the main text while a sentence has been added to qualitatively describe the formule for such indices.

– **Only first order indices can be estimated for multidimensional outputs? In Figure 10 I see that also the total indices are calculated at the landscape scale. How was this done ?** The formulation from previous Eq. (17) can actually be applied to Sobol' indices from any order. We have clarified the text and have explicitly mentioned the calculation of first and total order indices in Section 2.6.

– **Equation (24): If Xi and Y are not independent, the value of the dependence measure estimated for a given boot-strap resample (that is in a way obtained by randomly attributing values of Y to each value of Xi, if I understand correctly) will tend to be larger than the dependence measure estimated for the original non-bootstrapped sample? Why?** First, yes a bootstrap resample is indeed obtained by randomly attributing values of Y to each value of Xi. However, if Xi and Y are not independent, the HSIC value for such a bootstrap resample will be lower than the HSIC value for the original sample because therandom resampling step breaks the existing dependence relationship. The p-value then will tend to zero.

- **Section 2.4: The GSA workflow is not well explained in the text. In particular, the references to the sample sizes used are confusing. I read that 1000 points are used for PCE (L382), 4000 points for HSIC (L391), that 1000 points were derived from the 4000 points used for HSIC and that 1000 points are used for RF. It is only by looking at Figure 5 that I finally understood that these numbers are linked: 4000 points initially used for HSIC and then based on HSIC screening 1000 points are selected for all subsequent analyses. However, I am still a bit unsure why it is written L374 that 'a variance decomposition method was first used', isn't it HSIC?** First, a screening test is performed based on the statistical using HSIC from a 4,000-point LHS. Once influential parameters have been identified, a new 1,000-point LHS is generated with only influential parameters. On this new sample, Sobol, HSIC and RF indices are compared for ranking. This description has been explicitly integrated at the begining of Section 2.4, when merging Section 2.3 and 2.5. with clearer references to sample sizes.

- **L416 p17 '100 replications were used': Why using 100 replications forbootstrapping? 1000 bootstrap resamples are typically used (e.g. Archer et al., 1997; Yang, 2011).** Yes, indeed, we are aware that 1000 is a typical value for bootstrap resamples. However, such value was not affordable for estimating HSIC measures in a reasonable computing time. We then preferred to use 100 replications for all the tested methods, even the ones with low computational cost. Justification for this value has been added in the text.

- **Table 2: I believe that the LAImin and LAIharv are missing. The Table would also need to include an additional column that specifies at which spatial level the parameters are defined (e.g. soil horizon, plot/VFS). It took me a while and a bit of digging in the manuscript to get this information. I would also add the value of the standard scenario in Table 2, this would further improve readability.** As suggested, we added a column to Table 2 with spatial level definition and we also specified the values for the nominal simulation.

- **Section 2.5: this section does not clearly explain that the vegetation parameters and hpond are considered for vineyard plots and VFSs separately. As already mentioned in my previous comment, I think that the parameter should be clearly introduced in Section 2.1, which would improve readability and clarity.** Yes, modified as suggested

- **Section 3: As mentioned in my main comments, the manuscript lacks a discussion of the methodology and results with respect to previous studies, which could be highlighted in an additional discussion section.** As suggested, a discussion section has been added to comment on the global methodology and to put it into perspective in relation to previous studies.

- **P463 'It is commonly stated that [...]'. This sentence needs to be better justified. A reference is missing (e.g. Wagener & Pianosi, 2019). It can also be that many parameters are influential, but have only a small impact on the output except for a few parameters (e.g. five or six) that dominate the output variability.** Indeed,the sentence is inaccurate. The screening step intrinsically does not allow to draw conclusions on the number of parameters that dominate the output variability. We propose to eliminate the sentence to avoid confusion and hasty conclusions.

- **L566-568: Could you explain more why is it more costly to assess the sensitivity analysis at the local scale compared to the catchment scale? From Eq.17, it looks that anyway the catchment scale indices require the calculation of the local scale indices.** Indeed, in this case study we re-use the local scale indices to calculate the aggregated ones implying in this case no difference in computational cost. However, in its paper Gamboa et al. (2014) proposes an estimator for these aggregated indices that does not need the calculation of local indices. As local indices were calculated anyway in our case, we did not try such estimator but we mention it in the text since it seems very interesting to us, in the case the user does not want to compute local indices but directly the aggregated ones.

**Reviewer 2**

Thank you very much for the careful review and edits to the initital submission. Below we address the comments raised and we have also revised the manuscript accordingly to accommodate these.

**Main comments**

– **The novelty of this research needs more emphasis since the methods and algorithms are not new and the application of global sensitivity analysis in complex large-scale model is also not new (see Dai et al., 2017).** Indeed, the methodology we follow to perform sensitivity analysis in this study is a classical approach: first, a screening step and second, a ranking step applied on the reduced set of input parameters. However, the combined specificities of the application (high number of input parameters and high computational cost of a PESHMELBA simulation) are very limiting to perform each step. Alternative approaches that require limited sample sizes and that have been, up to now, poorly applied to pesticide model analysis are then necessary both for screening and ranking. In addition, combining several ranking methods with different definitions of sensitivity to get a comprehensive overview of influential parameters is also new. A discussion section about the full methodology has been added to argue on these points and to emphasize the novelty of this research.

– **The reasons of doing comparison for these three different sensitivity analysis methods need more discussions. Some conclusions for differences of these three methods are too obvious (e.g., the Sobol can consider the interactions).** Rather than comparing, in this study, we assume that combining different sensitivity analysis methods with contrasted definitions of sensitivity allows for building a robust and comprehensive overview of influential parameters on complex variables. For instance, using the HSIC dependence measure may allow to identify parameters that are influential in other quantities than second moment. This approach may be of particular interest for the variables considered in this case study as they result from the interactions of various physical processus and might be bimodal or highly skewed. However, as implementing several methods may not be possible in every case studies, comparing these methods regarding information it provides, accuracy and ease of implementation may also help future users to choose the most adapted approach for their case study. This justification has been added to the begining of Section 2.4 and the full argumentation has been modified so as not to only considered comparison of methods but also to justify to combine them. In addition, conclusions on the differences (or the lack of differences) between them have been consolidated refering to the difference in the sensitivity definitions they provide.

– **The screening procedure is unclear, what methods were used? The standard procedure is to use the Morris method or other low computational cost sensitivity analysis methods.** In this study, the Morris method could not be used due to 1) the high number of input parameters that led to fuzzy visual clustering and 2) the computational cost of a simulation that prevented us from running a large number of trajectories (see discussion of the revised manuscript for references of several studies that showed that a large number of trajectory is necessary to get robust screening results). Instead we used a statistical test for independence based on the HSIC measure. Mention to screening based on statisitical test has been added at the begining of Section 2.4 while justification for not using the classical Morris is provided in the discussion section.

– **The description of aggregated sensitivity indices is ambiguous, and the advantage of using it is not convincing.** Justification for using such aggregated indices is mainly to provide a summary of the overall sensitivity, especially to better target calibration effort. Also, such aggregated indices can be directly estimated, without performing a local GSA on each landscape element. This way, they can provide a rough sensitivity indicator if sufficient computational budget for local indice computation is not available. Justification for using them has been clarified in the manuscript. Also, the proof of such aggregated indice formulation has been replaced by a qualitative description to improve clarity and readability.

---

## Referee Report (RR1)

The submitted manuscript entitled "How to perform global sensitivity analysis of a catchment-scale, distributed pesticide transfer model? Application to the PESHMELBA model." by Rouzies et al. is deserved to publish after minor revisions, as follows.

1. The caption for the tables are normally written above tables.

2. P 9, Line 170: In table 2,

3. P10, Line 178: 2.38E-5

4. P27, Conclusion: The first paragraph of this section needs revision. It is mentioned "first", but, "second" and "third" are not come. Also, in Line 558, long and unclear sentence is written. In my point of view, first paragraph is not needed. Now, it is like a summary. The purpose is to focus on main messages and conclusions.

---

## Author Response (AR2)

**1 Reviewer 1**

Dear Reviewer 1,

Thank you for this new review. The manuscript has been extensively revised to take into account the comments of Reviewer 3 and we hope that it still meets your original requirements. In what follows, we address your remaining questions and comments. To be noted that some of them refer to parts of the manuscript that have been removed in this new version. Minor edits have been directly incorporated to the manuscript.

**I have a remaining comment regarding the input samples. The authors created an initial LHS sample of size 4000 to screen the influential parameter. Then they ranked these influential parameters based on a new sample of size 1000. Why is it required to create a new sample for the ranking? Could not the initial sample of size 4000 be used for this purpose (only considering the influential input and dropping the dimensions corresponding the non-influential inputs)? I think this should be clarified in the manuscript, since creating a new sample largely increases the computational cost of the analyses.**
Indeed, this strategy could be applied to limit the computational budget, especially since dropping the dimensions of some inputs would not affect the LHS structure of the sample. In this study, as the scenario was of limited size we could afford additional simulations and we generated new samples of different sizes to keep them as independent as possible, especially for the convergence study (that has been added to this new version). But your comment is fully relevant and we have added it as a perspective for a catchment-scale application that may be much more computationally costly.

**p3 L60-61 "Such approach [. . . ] information on the input.": this sentence if not clear and needs reformulation. Do you mean that this is a GSA method that can be applied 'given data', and does not require a specific structure for the input sample (see e.g. Saltelli et al., 2021)?**
Indeed, this sentence meant that the method does not require an input sample with a specific structure nor information about the input distributions. This specific sentence has been removed but the argument in favor of given-data methods now appears in the discussion part so as the reference to Saltelli et al., 2021.

**p3 L67: The model used in Vanuytrecht et al. (2014) is not a pesticide model, is it?**
Indeed it is not and the reference has been removed.

**p3 L69-70 "This qualitative method is based [. . . ] to make clusters appear.": references are missing to support this statement. For instance, Kim et al. (2022, Sect. 4.5) discusses the difficulties in applying Morris method in high dimensions.**
Thank you for the valuable reference. However, in the new version of the paper, most of the state-of-the-art on screening, including this sentence, has been removed. Indeed, we have decided to clearly focus the paper on the ranking step so as to provide more results and clearer arguments to choose a specific method.

**p11 L239 'accuracy': do you mean precision/robustness (assessed using bootstrapping)? I do not think from your analysis you can infer whether sensitivity indices are accurate. Can we know the 'true' value of the sensitivity indices?**

Indeed, we refered to robustness and precision when using the term 'accuracy'. However, I agree that the analysis provided in the initial version did not bring sufficient information to conclude about robustness nor precision of the indices. The new version of the paper now includes results on convergence of the calculated sensitivity indices and the associated error bounds so as to get clear conclusions on these aspects.

**p11 L254 "accurately": I would remove this term which I think may be misleading Sobol' indices are typically calculated numerically and not analytically. Can we be sure that the numerical procedure produces accurate sensitivity indices estimates?**
Yes, removed as suggested.

**p19 L454-455 'It may indicate [. . . ] output variance.': This sentence needs reformulation.**
This sentence does not appear anymore in the new version.

**p24 L548: I think this sentence needs clarification. The term 'relevancy' is vague. The term 'confidence' is a strong word and I do not think it is appropriate here, given the limitation of these methods discussed in the results section.**
The discussion on the methodology has been deeply reviewed and more precise arguments about 'relevancy' and 'confidence' on the methods have been proposed in this section.

**p25 L551-552: To put this into context, I suggest to refer to Saltelli et al. (2021), who highlights the benefit of 'given data' sensitivity analysis.**
Yes, added as suggested.

**2 Reviewer 3**

Dear Reviewer 3,

Thank you very much for the careful review and edits to the initital submission. All your comments and questions have been copied hereafter in bold then answered. the revised manuscript is provided as a complement to these answers.

**Overall evaluation: The submitted manuscript entitled "How to perform global sensitivity analysis of a catchment-scale, distributed pesticide transfer model? Application to the PESHMELBA model." by Rouzies et al. applies three GSA methods to evaluate the sensitivity of the distributed process-based model to its parameters. The writing is clear and precise, and all sections are understandable. Considering the importance of such analysis for complex hydrologic models, I think the motivation and benefits of this study will be of interest to Geoscientific Model Development readers. Particularly, I like the fact that various GSA methods have been compared in this paper. That being said, the manuscript suffers from some major shortcomings with respect to its novelty and rigor. Here, I outline my comments and suggestions that should allow authors to improve their paper:**

**Comment 1. The major shortcoming of the paper is that the overall value of this contribution to the hydrologic modelling community is not adequately discussed. The**

main contribution of this study is applying three GSA techniques to investigate the role of various parameters in pesticide transfer model. However, its merit over previous attempts is still somehow limited/not well presented. As mentioned by the authors, there are several studies where GSA approach has been applied to explore the factor importance in this context. I am not sure if this and similar studies would add much useful information to the existing body of knowledge on uncertainty analysis, parameter estimation, identifiability analysis, etc. I strongly suggest authors to clearly explain the extend to which this study is adding to the previously presented knowledge in the field (e.g., through new approaches to solving existing problems? etc.).

Having discussed the issue from that point of view, I would rather look at it from another perspective as well. Based on the reported results (Figure 7), overall, the estimated sensitivity indices by RF, HSIC, and Sobol methods are quite different. But, it is not convincing from the paper why one should use HSIC instead of Sobol or RF method. The manuscript correctly mentions the conceptual differences between three methods. For example, HSIC assesses the strength of dependencies between inputs and the output, while Sobol method attributes the variance of the output to variations in inputs or sets of inputs. However, it has not been discussed how this can help modelers/hydrologists with respect to hydrological processes' understanding or model building. To address this issue, I strongly suggest authors provide their "objectives" and "research questions" in the introduction section by bullet points. This can properly highlight the novelty and significance of the study. Furthermore, considering the numerical results, authors should explicitly explain why and how each GSA method might be useful in the context of spatialized pesticide transfer modelling.

As suggested, the objectives and the research questions of the paper have been precised as follows. The main focus of this study is to perform sensitivity analysis of the pesticide transfer model PESHMELBA in order to investigate the role of various parameters in the model. In the previous studies reported in this field, the Sobol method is commonly used for ranking. However, in such studies the number of parameters for ranking is limited (<25) while the number of simulations available is quite high (>10,000). In the case of PESHMELBA model and of similar distributed, multi-processes models, the structure and the spatialized aspect of the model imply a high number of input parameters ($\approx 150$) and a very limited number of available simulations (<5,000) due to their high computational cost. These constraints imply that classical approaches for GSA cannot be applied as it is to the PESHMELBA model. The objective of this study is then to identify an adequate approach that suits PESHMELBA constraints to perform global sensitivity analysis. The novelty of this study is that we test several low computational budget methods that have been very little used before to perform GSA of pesticide transfer models. The relevancy of these new methods is assessed regarding the following aspects :

- which information do they bring about sensitivity and more generally about physical processes envolved in pesticide transfer and transformation ?

- how robust are they in the case of small sample size ?

The introduction has been deeply reviewed to highlight the novelty of the study and the objectives and research questions as formulated above.

In addition, the discussion section has also been reviewed in order to draw clear conclusions about why and how each method might be useful. Having clarified the objectives of the study, the HSIC measure is particularly examined regarding the criteria above. As you suggested, its interest to help modelers with respect to hydrological processes is now properly discussed and we particularly

highlight the fact that its lack of interpretability is a significant drawback of such a GSA approach.

**Comment 2. In my opinion, another major shortcoming of this paper is that there is no information about the convergence behavior of the GSA algorithms. As authors know, robust sensitivity analysis of the models typically requires many model runs, and hence considerable computational resources. So, due to the high number of model evaluations required by existing sensitivity analysis techniques and the computationally expensive nature of the models, analysts usually tend to conduct sensitivity analysis without evaluating its stability and convergence (for more discussion see, e.g., Sarrazin et al., 2016; Sheikholeslami et al., 2020). It is therefore common to choose the sample size only based on the available computational budget, which in turn can result in lack of robustness, and consequently contaminate the assessment of the sensitivities. In fact, since 5 10 years ago a surge of papers flooded the environmental modelling journals introducing/applying a sensitivity analysis technique to a model without analyzing the robustness and converges of the results. Authors should properly monitor/analyze the convergence properties of the utilized GSA techniques in identifying influential factors, for example by progressively increasing the sample size.**

I definitely agree with you. As suggested, results about convergence properties of the tested methods have been added (Section 3.2.3) by progressively increasing the sample size up to the maximum possible sample size compatible with our computational budget (2,000 points for each variable for ranking). As already mentionned in the response to comment 1, convergence properties have been highlighted as a criterion for choosing the most suitable method for PESHMELBA.

**Comment 3. There is another important cost-effective strategy in the literature to accelerate GSA of the computationally expensive models, namely given-data approach to GSA (otherwise known as data-driven methods). To improve the literature review and strengthen the discussion part, authors can mention given-data approach in the revised manuscript. For a general review and discussion on these techniques see Sheikholeslami et al. (2021).**

If I understood correctly the definition of data-driven methods provided in Sheikholeslami et al. (2021), all the methods we applied in this study are data-driven methods as they do not require a specific sample design. Such an argument in favor of these methods has been added in the discussion part.

**Comment 4. Going back to comment 3, I think an insufficient state-of-the-art has been performed in this study. There are many studies that have been previously undertaken to develop efficient screening techniques. Authors should consider existing literature in this context and perform a critical review. One notable example is the grouping approach introduced by Sheikholeslami et al. (2019). This approach uses agglomerative hierarchical clustering to categorize the parameters into distinct groups based on similarities between their sensitivity indices, and then ranks parameters according to importance group e.g., these could be labeled as "strongly influential", "influential", "moderately influential", "weakly influential", and "non-influential") rather than individually (see Huo et al., 2019; Sheikholeslami et al., 2021 for further application of the grouping-based importance ranking approach). Other studies include Tang et al., 2007; Nossent et al., 2011; Touzani and Busby, 2014; Becker et al., 2018; etc.**

In this study, screening was only performed as a preliminary step to get a reasonable number of input parameters for ranking. We intentionally kept the part of screening short because the focus of

the paper is rather on ranking. To avoid confusion, we have further shortened this part to keep the paper reasonably short. However, the discussion section now mentions that further research should specifically target the screening exercise, including by exploring the approaches you cited.

**Comment 5. While "parameter uncertainty" has been thoroughly analyzed in the paper, I could not find proper description about other important sources of uncertainty, particularly input data uncertainties, e.g., soil type, rain and PET forcing data. I recommend authors to add discussion regarding this important source of uncertainty which can significantly affect the model output variability. In fact, these forcings are typically the outputs of a long and complex modelling chains. Thus, PESHMELBA may simultaneously suffer from model parameter, model structure, and input uncertainties or other systematic uncertainties in precipitation bias correction, the estimation of potential evapotranspiration, or the uncertainty of deriving spatial basin scale meteorological input data.**
Yes, added as suggested. We are aware that parameter uncertainty is not the only source of uncertainty that can affect the model but it is the only one that is considered in this study for sake of simplicity. Discussion about other sources of uncertainty such as data and model structure have been added as a perspective in the conclusion.

**Comment 6. Most of the existing literature on sensitivity analysis has typically been under the assumption that the controlling factors such as model parameters (processes) are independent, whereas, in many cases, they are correlated, and their joint distribution follows a variety of forms. However, very few studies in the field of water and environmental modeling address this issue. By way of example, Strobl et al. (2007) reported that when using permutation-based mean decrease in prediction accuracy as an importance measure, there might be bias in estimating importance of the correlated variables. Authors should highlight this in the revised manuscript by adding discussion on the significance of correlation effects in the utilized methods and then perhaps propose strategies (in future studies) for properly accounting for correlations in parameter (process) space.**
This is a very interesting and relevant remark and we added as a perspective in the conclusion. To keep it synthetic, we mainly mentioned the fact that the indices we used in this study may be meaningless in the case of dependent inputs. The main strategy we propose to deal with dependent inputs consists in using the Shapley effects as this lead is very promising and can be adapted to any type of sensitivity indice (see Da Veiga et al. 2021 for further details).

**Comment 7. I could not find any information on training and tuning of RF model. The possible inconsistency in SA results might be due to the issues in fitting RF to the input-output data. I strongly suggest authors provide details of building RF model. Furthermore, it's not clear if RF was fitted on scalar quantities or temporal series. Without this information, results are not reliable and cannot be validated.**
Details on RF building are provided l.349 and l.350 : The randomForestSRC R package (Ishwaran and Kogalur, 2020) was used to train RF and the number of trees used for training was set to 500. As suggested, we also clarified the fact that the variables used for training were all scalar quantities and that one RF was trained for each variable on each plot.

**Comment 8. It would be interesting to see results of parameter ranking as well. Although these methods estimate different values for sensitivity indices in some cases,**

**the ranking provided by these methods may be much more similar. Note that, in complex models, when the number of parameters is very large, we are typically not interested in an exact values of sensitivity indices. Instead, it may be more profitable to use the available computational budget to rank parameters in order of importance, e.g., "strongly influential", "moderately influential", and "non-influential".**

In this case, we are interested in providing a quantified ranking rather than an agglomerative hierarchical clustering because a side objective is to focus parameter characterization efforts. As suggested we have added some results about parameter ranking together with comparison of rankings (Figure 6) and convergence monitoring (Figures 7 and 8).

**Comment 9. A possible direction for future research is to evaluate how sensitivity analysis results change by changing the selected parameter distributions (normal, log-normal, uniform,...) since there is an unavoidable uncertainty associated with defining feasible ranges of parameters.**

It is indeed a possible direction for future research but we chose to prioritize the question of dealing with dependent inputs in the conclusion part as it has turned a burning question in the community of sensitivity analysis during the last years.

---

## Author Response (AR4)

Dear Reviewer 4,

Thank you for this new review. The manuscript has been corrected in order to address your comments :

1. All captions have been moved above the tables and figures.
2. P9, line170 : ok, coma added as suggested
3. P10, we suggest to keep the scientific notation as it is in order to be consistent with Table 2 and Table 3.
4. P27, the conclusion has been modified as suggested. The first paragraph has been deeply shorten in order to focus on take-home messages.